# Downscaling Multi-Model Climate Projection Ensembles with Deep Learning (DeepESD): Contribution to CORDEX EUR-44

Jorge Baño-Medina[1], Rodrigo Manzanas[2,3], Ezequiel Cimadevilla[1], Jesús Fernández[1], Jose González-Abad[1], Antonio S. Cofiño[1], and José Manuel Gutiérrez[1]

[1]Instituto de Física de Cantabria (IFCA), CSIC-Universidad de Cantabria, Santander,Spain
[2]Dept. Matemática Aplicada y Ciencias de la Computación (MACC), Universidad de Cantabria, Santander, Spain
[3]Grupo de Meteorología y Computación, Universidad de Cantabria, Unidad Asociada al CSIC, Santander, Spain

**Correspondence:** Jorge Baño-Medina (bmedina@ifca.unican.es)

**Abstract.** Deep Learning (DL) has recently emerged as an innovative tool to downscale climate variables from large-scale atmospheric fields under the perfect prognosis (PP) approach. Different Convolutional Neural Networks (CNN) have been applied under present-day conditions with promising results, but little is known about their suitability for extrapolating future climate change conditions. Here, we analyze this problem from a multi-model perspective, developing and evaluating an ensemble of CNN-based downscaled projections (hereafter DeepESD) for temperature and precipitation over the European EUR-44i ($0.5°$) domain, based on eight GCMs from the Coupled Model Intercomparison Project Phase 5 (CMIP5). To our knowledge, this is the first time that CNNs have been used to produce downscaled multi-model ensembles based on the perfect-prognosis approach, allowing to quantify inter-model uncertainty in climate change signals. The results are compared with those corresponding to an EUR-44 ensemble of regional climate models (RCMs) showing that DeepESD reduces distributional biases in the historical period. Moreover, the resulting climate change signals are broadly comparable to those obtained with the RCMs, with similar spatial structures. As for the uncertainty of the climate change signal (measured on the basis of inter-model spread), DeepESD preserves the uncertainty for temperature and results in a reduced uncertainty for precipitation.

To facilitate further studies of this downscaling approach we follow FAIR principles and make publicly available the code (a Jupyter notebook) and the DeepESD dataset. In particular, DeepESD is published at the Earth System Grid Federation (ESGF), as the first continental-wide PP dataset contributing to CORDEX (EUR-44).

## 1   Introduction

The Coupled Model Intercomparison (CMIP) initiative produces periodic multi-model ensembles of centennial global climate projections under different future scenarios using Global Circulation Models (GCMs). The two latest ensembles available are CMIP5 (Taylor et al., 2012) and CMIP6 (Eyring et al., 2016), with typical resolutions of around 200 and 100km, respectively. These results are widely used by the impacts and adaptation communities in different sectors (e.g., energy, agriculture and health, among others). However, the biases and spatial resolution of these global projections hamper their use in regional applications and different downscaling approaches and methods are routinely applied to produce actionable information at the regional and local scales (Maraun and Widmann, 2018).

Dynamical downscaling is based on the use of Regional Climate Models (RCMs) over a limited region driven by GCM outputs at the boundaries (Giorgi, 2019; Gutowski et al., 2020). Different regional initiatives provide high-resolution, physically consistent downscaled simulations over continental-wide domains. In particular, the Coordinated Regional Climate Downscaling Experiment (CORDEX, https://cordex.org) provides multi-model ensembles of regional climate projections driven by CMIP5 model outputs over 14 continental domains. These regional projections are highly demanding in terms of computational resources and the resolution of the available regional projections ranges from 50 to 10km, depending on the domain.

The empirical-statistical downscaling approach (ESD) is based on empirical/statistical models translating the coarse-resolution information provided by the GCMs (*predictors*) to the regional/local scale provided by the available historical observations (*predictands*), typically temperature or precipitation fields (Gutiérrez et al., 2019). Under the "perfect-prognosis" (PP) approach, the statistical models are trained in a historical period to learn a predictor-predictand link using simultaneous observed and reanalysis (quasi-observations) values (daily in this work) for predictands and predictors, respectively. The resulting models are then applied to GCM predictor values (from present climate or future scenarios) to obtain the regional/local downscaled results. This approach is based on a number of assumptions. For example, predictors have to be realistically simulated by GCMs (e.g. exhibiting small systematic biases), so large-scale fields in upper levels (less affected by orography and model resolution) are typically used as predictors (perfect prognosis assumption); moreover, the statistical models trained in present climate conditions should remain valid under modified (out-of-sample) climate conditions (generalization assumption) (see Gutiérrez et al., 2019, for more details). Compared to dynamical downscaling, ESD lacks explicit physics in the model formulation and typically does not ensure full multivariate (intervariate and spatial) consistency. However, these methods overcome the systematic biases present in RCM products (as the model is trained using observations) and are not computationally demanding, avoiding the need for large computational infrastructures (Le Roux et al., 2018). Therefore, these methods could be widely used to downscale global multi-model ensembles providing results at continental scales, e.g. in CORDEX domains.

Recently, deep learning methods based on Convolutional Neural Networks (CNNs) have become very popular as a statistical downscaling technique due to their ability to achieve an automatic selection of predictors in the form of data-driven spatial features (Baño-Medina, 2020). Although they have shown promising results for continental-level climate downscaling under "perfect" conditions (Pan et al., 2019; Baño-Medina et al., 2020; Sun and Lan, 2021; François et al., 2021), there is little knowledge on whether these statistical models are able to generalize to out-of-sample climate change conditions. Some preliminary work using a single GCM shows that CNNs can accurately reproduce the local climate variability and provide plausible climate change projections over Europe as compared to well-established statistical downscaling approaches (Baño-Medina et al., 2021). However, further analysis along these lines is needed to assess the suitability of CNNs for climate change applications.

Here we provide a multi-model perspective by applying a CNN model (Baño-Medina et al., 2021) to downscale daily precipitation and temperature over Europe from the historical and future projections (RCP8.5 scenario) provided by an ensemble of eight GCMs. We evaluate the consistency of the downscaling approach across models and analyze the uncertainty of the resulting climate change signals. Moreover, we follow previous downscaling literature (Vrac et al., 2007; San-Martín et al., 2017; Quesada-Chacón et al., 2021) and compare the resulting projections with an ensemble of RCMs, which are used as "pseudo-observations". In order to facilitate further analysis, this dataset (referred to as Deep learning Empirical Statistical

Downscaling (DeepESD)) is made publicly available on the Earth System Grid Federation (ESGF), as a contribution to the EUR-44i domain (0.5° horizontal resolution), so it can be downloaded together with the ensemble of available RCMs. To our knowledge, this is the first continental-scale climate change projection dataset produced using statistical downscaling methods contributing to CORDEX and published in ESGF, following the standard procedure for RCMs. Moreover, following FAIR principles (Wilkinson et al., 2016), the code used to generate the dataset along with guidelines on how to access the data is available on Zenodo (see the section on code and data availability).

## 2 Data and Methods

Following the PP approach, the CNN models have been trained over the period 1979-2005 using daily predictors from the ERA-Interim reanalysis (Berrisford et al., 2011), upscaled from its original 0.75° resolution to a reference 2° regular grid, and predictands from E-OBS v20 (Cornes et al., 2018), originally at 0.25° but upscaled to 0.5° for consistency with previous works (Baño-Medina et al., 2020; Baño-Medina et al., 2021). E-OBS is a high-resolution observational dataset generated by spatially interpolating the European Climate Assessment & Dataset (ECA&D) network of stations (Klok and Klein Tank, 2009). Although national and sub-national datasets exist, E-OBS accurately represents the regional climate over the entire European continent (Bandhauer et al., 2022) and it is commonly used in continental-wide statistical downscaling experiments (Maraun et al., 2015; Vrac and Ayar, 2016; Baño-Medina et al., 2020; Baño-Medina et al., 2021). We chose version 20 (v20, release date October 2019) since it was the most recent at the start of this study. Following previous studies (Gutiérrez et al., 2019; Baño-Medina et al., 2020), air temperature, specific humidity, geopotential, meridional and zonal wind velocity at 500, 700, 850 hPa (i.e., a total of 15 variables per gridbox) have been used as predictors covering the domain $34°N$-$76°N$, $8°W$-$34°E$, resulting in a $22\times22\times15$ (longitude$\times$latitude$\times$variable) high-dimensional input grid. To avoid potential artifacts derived from the different scale of the distinct variables, ERA-Interim predictors are standardized at the gridbox level (Baño-Medina et al., 2021).

For downscaling we used an ensemble formed by the eight CMIP5 GCMs described in Table1, whose ability to reproduce the large-scale dynamics has already been assessed for PP studies (Brands et al., 2013) and have also been used in EURO-CORDEX to drive RCMs (Vautard et al., 2021). Therefore, we apply our trained models to downscale the projections from the this ensemble for the historical (1975-2005) and RCP8.5 scenario (2006-2100) periods. We follow previous work in this field (Baño-Medina et al., 2021; Olmo et al., 2022) and select the RCP8.5 scenario, which shows the strongest climate change signal (especially for temperature) and therefore allows the generalization capability of CNNs to be optimally explored. Due to their different spatial resolutions, all GCM data have been interpolated to the reference 2° grid (considering the nearest gridbox) to match the predictor space used for ERA-Interim. No differences in the downscaled results were found by employing other interpolation techniques (e.g., bilinear). Moreover, in order to reduce potential systematic biases in GCM predictors which may affect the perfect prognosis assumption, we bias adjust GCM predictors towards the corresponding reanalysis values. As suggested in previous studies we use a change-preserving method (Vrac and Ayar, 2016) in order to avoid introducing artificial trends/changes in future GCM predictor values. In particular we use a simple scaling method (mean and variance) applied at a

| Name | Institution | Spatial resolution |
|---|---|---|
| CanESM2 (Christian et al., 2010) | Canadian Centre for Climate Modelling and Analysis | $(2.81°, 2.79°)$ |
| CNRM-CM5 (Voldoire et al., 2013) | Centre National de Recherches Météorologiques and Centre Européen de Recherche et de Formation Avancée | $(1.4°, 1.4°)$ |
| MPI-ESM-MR (Müller et al., 2018) | Max-Planck Institut für Meteorologie | $(1.87°, 1.87°)$ |
| MPI-ESM-LR (Müller et al., 2018) | Max-Planck Institut für Meteorologie | $(1.87°, 1.87°)$ |
| NorESM1-M (Bentsen et al., 2013) | Norwegian Climate Center | $(2.5°, 1.9°)$ |
| GFDL-ESM2M (Dunne et al., 2013) | National Oceanic and Atmospheric Administration Geophysical Fluid Dynamics Laboratory | $(2.5°, 2.02°)$ |
| EC-EARTH (Doblas Reyes et al., 2018) | European-wide consortium | $(1.12°, 1.12°)$ |
| IPSL-CM5A-MR (Dufresne et al., 2013) | Institut Pierre Simon Laplace Climate Modelling Center | $(2.5°, 1.27°)$ |

**Table 1.** The different CMIP5 models used in this study.

monthly scale; for future periods, the climate change signal is removed from the data before bias adjustment and added to the results. We want to remark that we tested both signal-preserving and standard bias adjustment obtaining substantial differences in the climate change signals for temperature; signal-preserving yields more plausible results (as compared with GCM and

95 RCM climate change signals). As in the case of the reanalysis, GCM predictors are standardized at the gridbox level for their use in the CNN (the same standardization parameters used for the reanalysis data is applied here).

The above pre-processing steps are illustrated in Fig. 1.

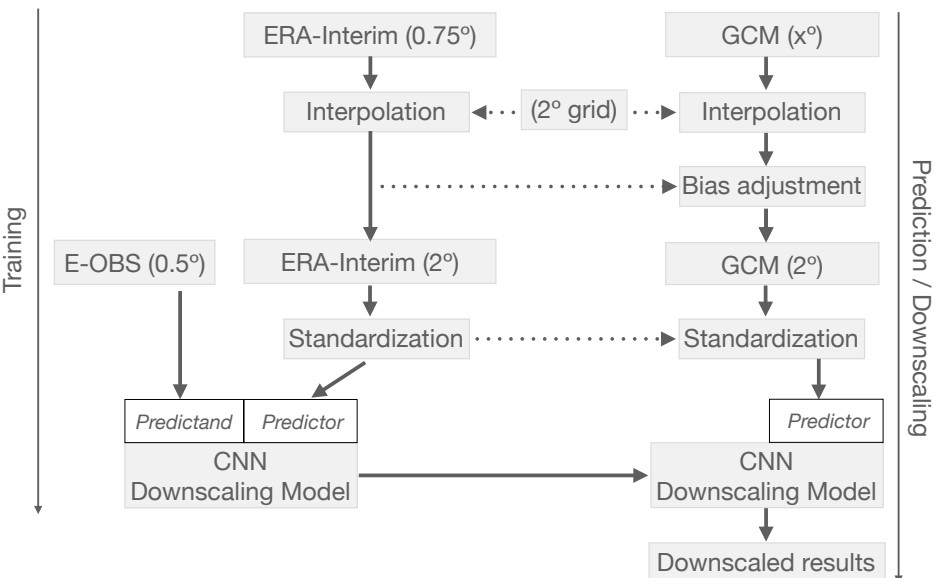

**Figure 1.** Workflow of pre-processing steps applied to reanalysis and GCM data in this work.

| GCM | member | RCM | Institution |
|---|---|---|---|
| CanESM2 | 1 | SMHI-RCA4 | Swedish Meteorological and Hydrological Institute, Rossby Centre |
| CNRM-CM5 | 1 | CLMcom-CCLM5-0-6 | Climate Limited-area Modelling Community |
| CNRM-CM5 | 1 | SMHI-RCA4 | Swedish Meteorological and Hydrological Institute, Rossby Centre |
| MPI-ESM-LR | 1 | CLMcom-CCLM4-8-17 | Climate Limited-area Modelling Community |
| MPI-ESM-LR | 1 | MPI-CSC-REMO2009 | Max Planck Institute for Meteorology |
| NorESM1-M | 1 | SMHI-RCA4 | Swedish Meteorological and Hydrological Institute, Rossby Centre |
| GFDL-ESM2M | 1 | SMHI-RCA4 | Swedish Meteorological and Hydrological Institute, Rossby Centre |
| EC-EARTH | 12 | SMHI-RCA4 | Swedish Meteorological and Hydrological Institute, Rossby Centre |
| EC-EARTH | 12 | CLMcom-CCLM5-0-6 | Climate Limited-area Modelling Community |
| IPSL-CM5A-MR | 1 | SMHI-RCA4 | Swedish Meteorological and Hydrological Institute, Rossby Centre |
| IPSL-CM5A-MR | 1 | IPSL-INERIS-WRF331F | Institut Pierre-Simon Laplace |

**Table 2.** Details of the EURO-CORDEX (EUR-44 domain) RCM simulations used in this study. The first two columns show the GCM and ensemble member driving the RCM.

For the CNN models used in this work, we deploy the best performing topologies developed in (Baño-Medina et al., 2020), a recent study which intercompares different CNNs over Europe to downscale temperature (precipitation). They consist of three convolutional layers (LeCun et al., 1995) with 50, 25 and 10 (1) spatial kernels (3x3 gridboxes) followed by a dense connection linking the last hidden layer to the output neurons (corresponding to the land gridpoints in E-OBS). As in (Baño-Medina et al., 2020) we apply a distributional downscaling approach and use the network to estimate daily predictor-conditioned Gaussian (Bernoulli-Gamma) distributions for temperature (precipitation). This is implemented for each land gridbox using two (three) output neurons corresponding to the distributional parameters: mean and variance (probability of rain, shape and scale factors). The resulting networks are trained to optimize the negative log-likelihood of the Gaussian (Bernoulli-Gamma) distribution. We refer the reader to Baño-Medina et al. (2020) for more details. During calibration, we use a test set (randomly selected 10% of the data) to perform early-stopping, and stop training when the test error stops decreasing after 30 epochs.

The computations performed in this work were executed on a single node 2x Intel(R) Xeon(R) E5-2670 0 @ 2.60GHz CPU (16 cores) with 60 GiB of RAM. The computational time taken to calibrate the model and generate the projections for a GCM was less than six hours, which is considerably less than the time required to run a similar experiment with an RCM (for instance, the EUR-44 simulations performed with the WRF model for a single GCM in Fernández et al., 2019, lasted six months using 9 nodes with 144 cores). This approach can provide either deterministic predictions, by considering the expected value of the distribution for each day and grid-point, or stochastic ones, by simulating a random value from the distribution. Note that the deterministic approach typically result in an underestimation of the variability (and the extremes), since the explained variance may be significantly smaller than the observed one (Williams, 1998; Cannon, 2008; Baño-Medina et al., 2020). This is especially relevant for precipitation, whose local variability is often influenced by local phenomena which are not captured by the chosen predictors (Schoof and Pryor, 2001; Maraun and Widmann, 2018). We analyzed both deterministic

and stochastic approaches and finally used the stochastic (deterministic) version of the precipitation (temperature) downscaled fields. For the stochastic version we tested the results for different realizations and found robust results for historical biases and climate change signals.

We use a set of CORDEX RCMs (EUR-44 domain, Table 2) to analyze the generalization to out-of-sample climate change conditions of the CNN-based regional projections. Using RCM simulations as "pseudo-observations" is a common procedure adopted in the literature to validate ESD downscaled projections for future scenarios (Vrac et al., 2007; San-Martín et al., 2017; Quesada-Chacón et al., 2021). Nevertheless, note that RCMs still suffer from deficiencies in their model formulations that may affect their futures estimates (Boé et al., 2020; Gutiérrez et al., 2020), and therefore they should not be considered as purely true values for the CNN projections but rather as plausible trajectories. For a direct comparison, we interpolate these RCMs from their original spatial resolution (0.44°) to the predictand 0.5° regular grid.

Finally, we test the sensitivity of CNN training on the results by repeating the downscaling experiment ten times and evaluate historical biases and future climate change signals as shown below without finding appreciable variations.

## 3   Results

Figure 2 shows mean daily precipitation and temperature over the historical period 1975-2005 (and biases relative to E-OBS) for the multi-model means provided by the GCMs, RCMs and DeepESD ensembles. For precipitation, the raw GCM results show a smooth spatial pattern which does not capture the strong local-to-regional variability of this variable, and both GCM and RCM overestimate rainfall over most of the domain. As expected, DeepESD exhibits a largely unbiased spatial pattern over the entire continent, which is a result of being trained directly with observations. For temperature all approaches capture the latitudinal gradient, but both GCM and RCM results exhibit important biases over vast regions of the continent with predominant negative biases for RCM results. Again, DeepESD yields a mostly unbiased spatial pattern as a consequence of the training process (Casanueva et al., 2016). Besides these results for the mean, Figure 3 compares the entire precipitation and temperature distributions for the GCM, RCM and DeepESD ensembles over the historical period 1979-2005, for three different illustrative regions (the Alps, Iberian Peninsula and Eastern Europe). The reduction of biases for DeepESD is noticeable along the entire distribution (including the extremes) for both precipitation and temperature. Note that for precipitation these results are due to the use of the stochastic nature of the method, sampling from the inferred conditional distributions.

Figure 4 shows the mean climate change signal resulting from the GCM, RCM and DeepESD ensembles, as well as the underlying uncertainty (characterized by multi-model dispersion). In particular, the right (left) panel in this figure shows the values for precipitation (temperature) for near, mid and far future periods (rows 1-3) relative to 1975-2005, as projected by the GCM, RCM and DeepESD ensembles (in columns).

Overall, the spatial pattern of future precipitation changes is similar for the three ensembles, with precipitation decreasing over Southern Europe and increasing over the Northern part of the continent. Slight regional differences exist among the three ensembles, with DeepESD presenting weaker (decreasing) signals of change over the Iberian Peninsula but stronger (increasing) ones over some parts of Northern and Eastern Europe, especially when compared with GCMs. Interestingly, both

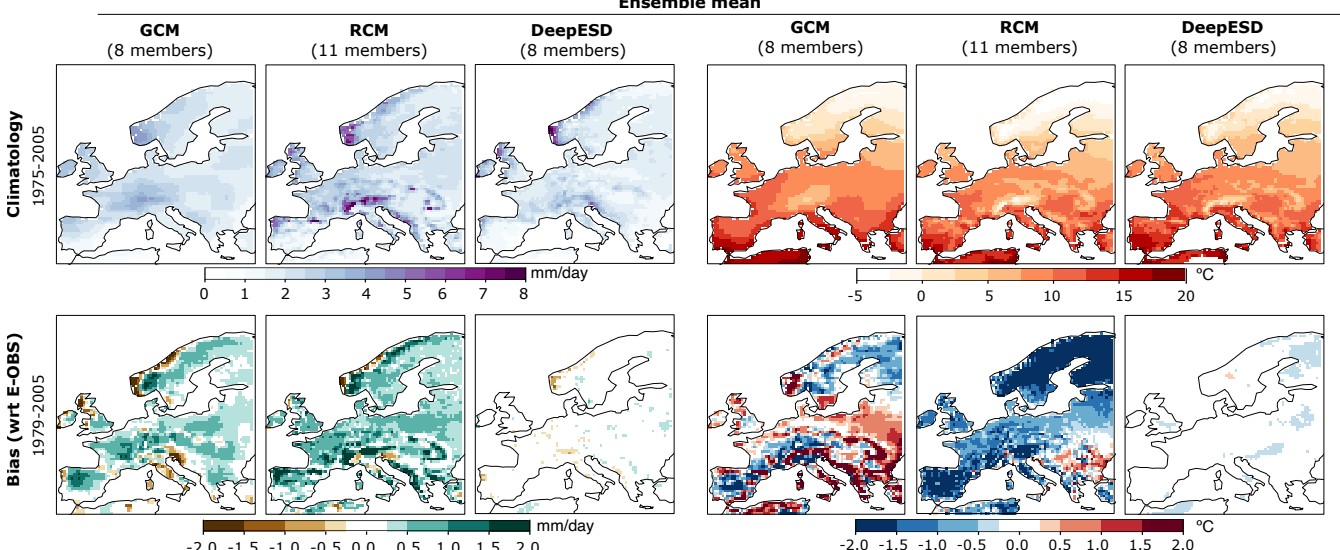

**Figure 2.** Annual daily precipitation (left block) and temperature (right) for the historical period 1975-2005, as obtained from the ensembles of GCMs, RCMs and DeepESD GCM-downscaled results (left, middle and right columns, respectively). The first row shows the ensemble mean climatological values and the second row displays the corresponding biases with respect to E-OBS v20.

the climate change signal and the multi-model uncertainty spatial patterns of DeepESD are more similar to the downscaled RCM than to the GCM ensemble. Moreover, DeepESD projects lower uncertainty than both physical-based ensembles across most of the European continent.

Regarding temperature, the spatial patterns are broadly consistent among the three ensembles, with the highest warming located over Northern Scandinavia, Eastern Europe and the Mediterranean basin, and the lowest one for the British Isles and Western and Central Europe. As in the case of precipitation, some regional differences exist among ensembles, especially over Central and Eastern Europe where both RCMs and DeepESD project lower signals of change than the GCMs, reducing the warming signal by about 0.5-1°C by the end of the century. Finally, the GCMs' ensemble spread range between 0.5-1.5°C with higher values in Southern and especially Northern Europe than in the rest of the continent. The RCMs (DeepESD) project a similar spatial pattern than the GCMs with a lower spread over Central and Eastern Europe (Scandinavia).

Further research is needed to assess whether the differences between GCM and RCM/DeepESD signals are due to an added value of downscaling or to deficiencies in the models. In the case of the RCMs, some recent studies attribute them to the lack of time-varying anthropogenic aerosols in the RCM formulation (Boé et al., 2020; Gutiérrez et al., 2020). To further analyze the results for DeepESD, Figure 5 shows (in columns) the climate change signals (2071-2100 with respect to 1975-2005) of the eight CMIP5 climate models considered and the corresponding DeepESD downscaled fields for precipitation (rows 1-2) and temperature (rows 4-5). Rows 3 and 6 show the differences between the DeepESD downscaled and raw climate change signals for the different GCMs. As per the climate change signal of precipitation, we observe a South-to-North gradient with different intensities depending on the GCM (i.e., MPI, GFDL and IPSL present the lowest values over the Mediterranean basin).

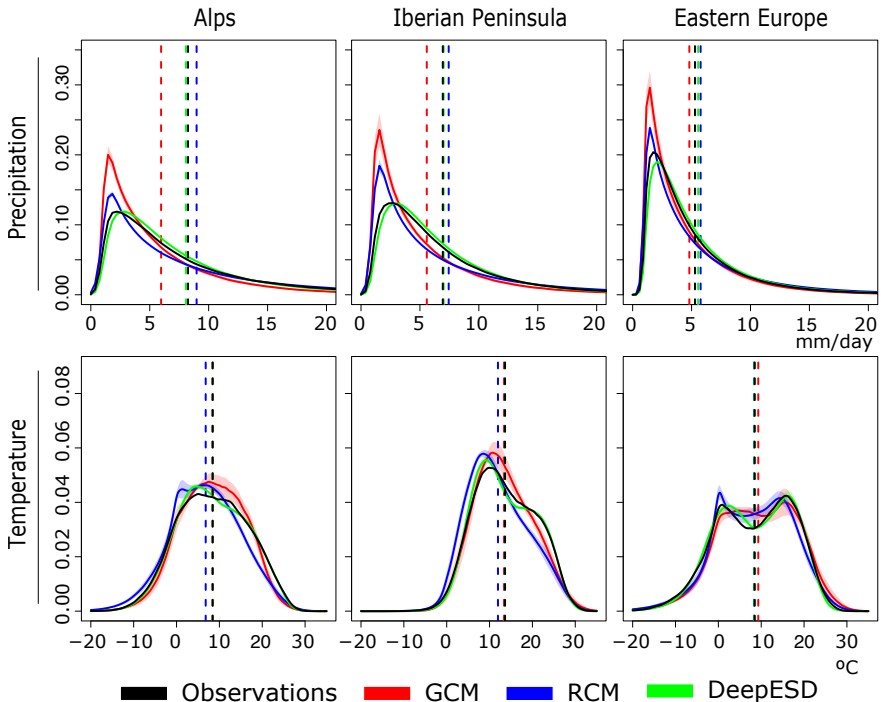

**Figure 3.** Probability Density Functions (PDFs) of the GCM (red), RCM (blue) and DeepESD (green) ensembles of precipitation and temperature over the historical period 1979-2005, plus E-OBS (black) for the Alps, the Iberian Peninsula and Eastern Europe as defined in the PRUDENCE regions (Christensen and Christensen, 2007). The solid line represents the ensemble mean and the shadow encompass two standard deviations. The dashed line indicates the distributional mean of each PDF.

Differently to the GCMs signal, DeepESD provide a more homogeneous spatial pattern explaining the low inter-model spread
of Figure 4. As per temperature, all GCMs project a positive climate change signal with subtle spatial patterns which vary across GCMs that are well captured by the DeepESD downscaled fields. This similarity in the climate change signals between the GCM and DeepESD fields explain the similar inter-model spread of Figure 4. Also, we observe that the CNRM-CM5 is the one model responsible for the reduced warming signal over Eastern Europe described in Figure 4.

To examine the behaviour of CNNs beyond climatological fields, Figure 6 shows the yearly time-series for precipitation and
temperature averaged over Eastern the Alps, the Iberian Peninsula and Eastern Europe domains, as defined in the PRUDENCE regions (Christensen and Christensen, 2007), which are broadly representative of the different European climate regimes — mountainous, mediterranean and continental, respectively.— Namely, we focus on the frequency of rainy days (R01), i.e. those receiving at least 1mm of rain, the average precipitation in rainy days (SDII) and the mean of temperature. For every indicator, the ensemble of GCMs (red), RCMs (blue) and DeepESD (yellow) for the total period 1975-2100, plus the observational
reference, E-OBS (black), for the period 1979-2008, are shown. In all cases, the solid lines represent the multi-model ensemble mean whilst the shadows encompass all the models contributing to the ensemble.

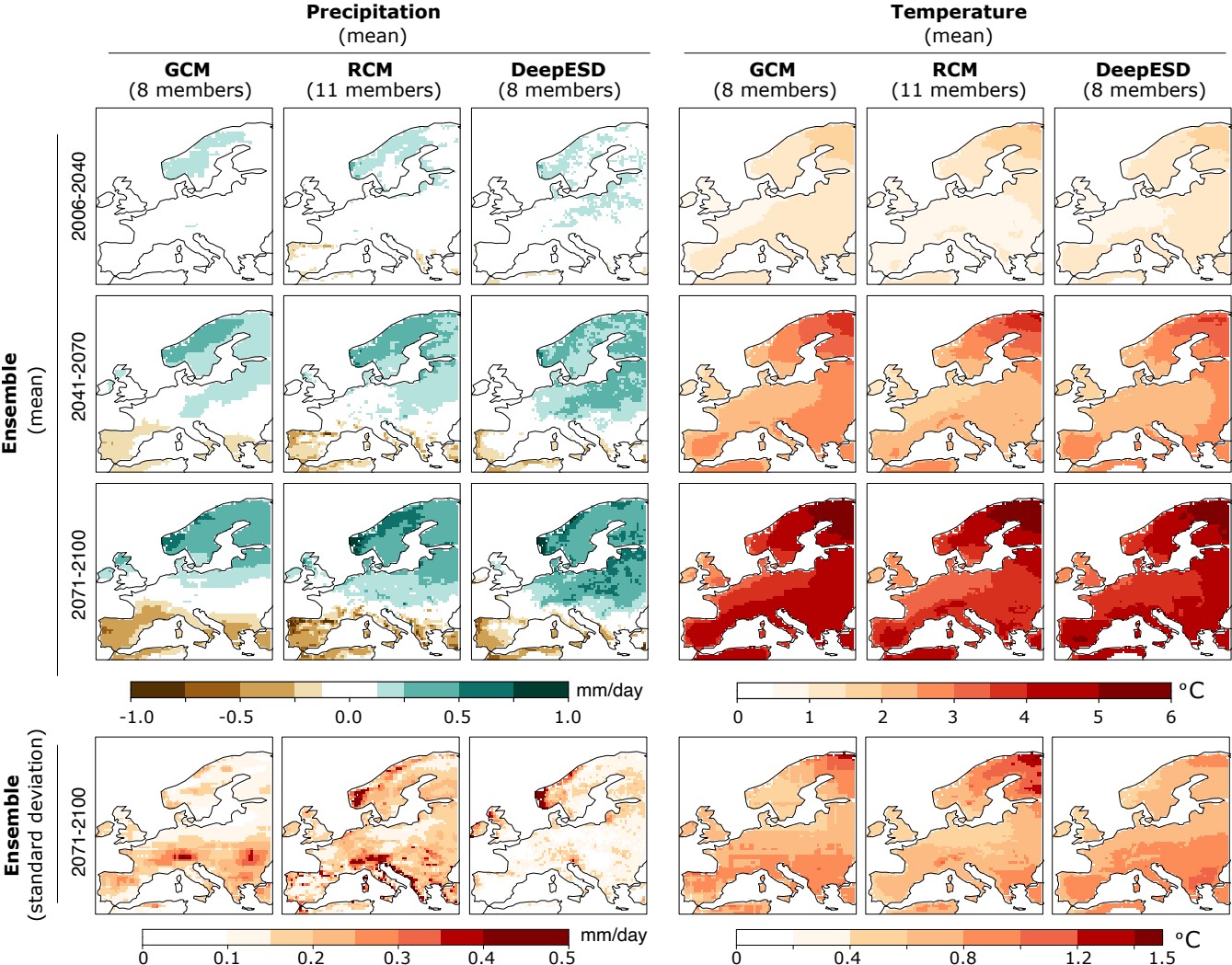

**Figure 4.** Climate change signal for annual mean precipitation (left) and temperature (right) for near- (2006-2040), mid- (2041-2070) and far-future (2071-2100) periods, in rows, relative to 1975-2005 as projected by the GCM, RCM and DeepESD ensembles (in columns). The last row shows the uncertainty of the far-future signal, as measured by the standard deviation of the results across models.

Figure 6.a shows that both GCMs and RCMs overestimate the frequency of wet days with respect to the observational reference —consequence of the drizzle effect (Dai, 2006).— For the SDII, RCMs present mostly unbiased fields whilst GCMs underestimate this metric across all regions, remarking the added value of RCMs to reproduce regional precipitation. In contrast to GCMs and RCMs, DeepESD provides in general more robust estimates for both R01 and SDII under the historical scenario. In terms of future changes, the three ensembles project an increase in in the SDII across all regions, and a decrease (increase) of the number of wet days in the Southern (Northern) regions consistent with the results of Figure 2.

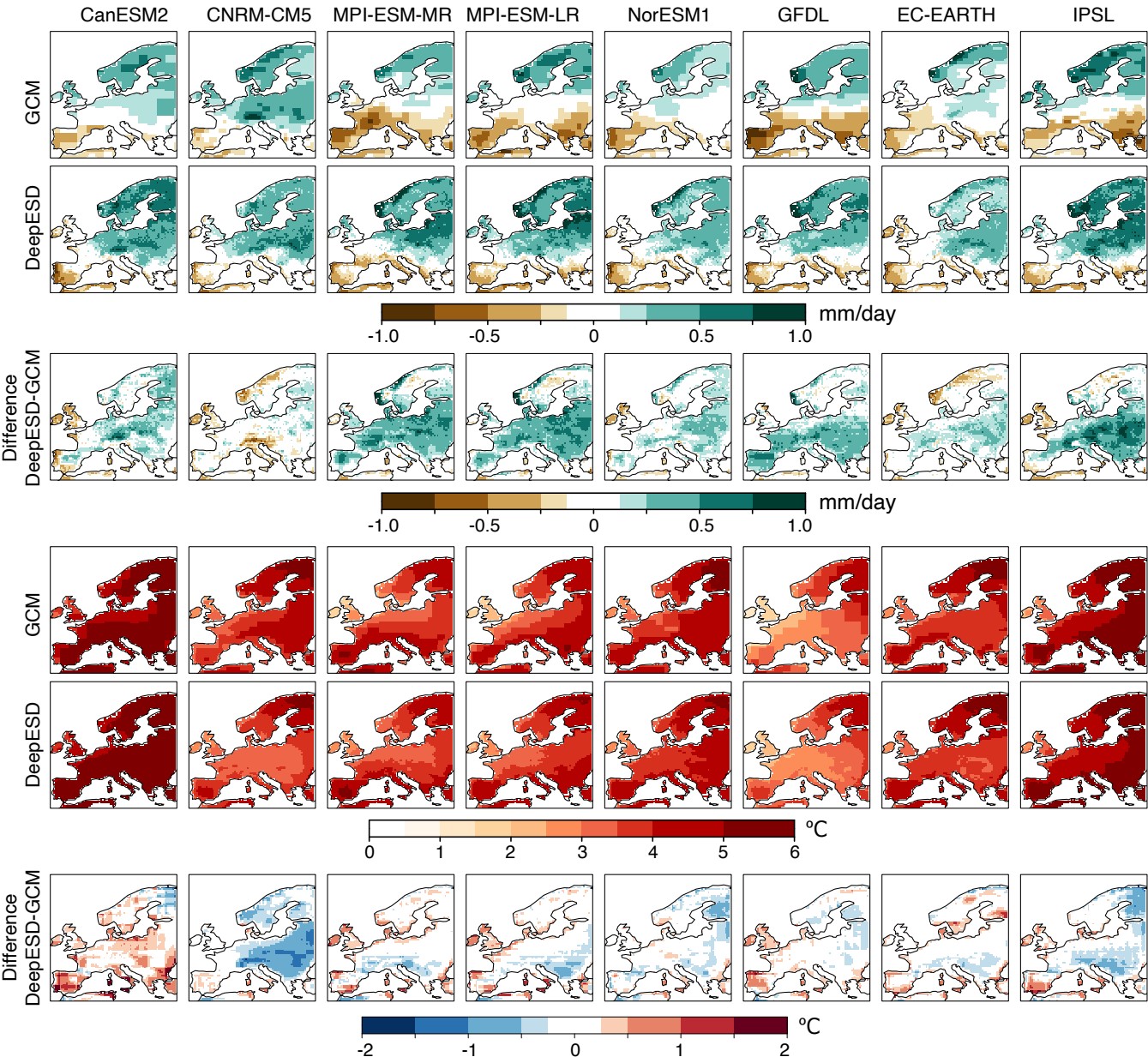

**Figure 5.** The climate change signals (2071-2100 with respect to 1975-2005) of the eight CMIP5 climate models considered and the equivalent DeepESD downscaled fields for precipitation (rows 1-2) and temperature (rows 4-5). Rows 3 and 6 show the difference between the DeepESD downscaled and raw climate change signals for the different GCMs

For temperature, Figure 6.c shows that the three ensembles perform similarly across all regions, with some systematic underestimation of mean temperatures by the RCMs, and DeepESD exhibiting nearly unbiased results under the historical

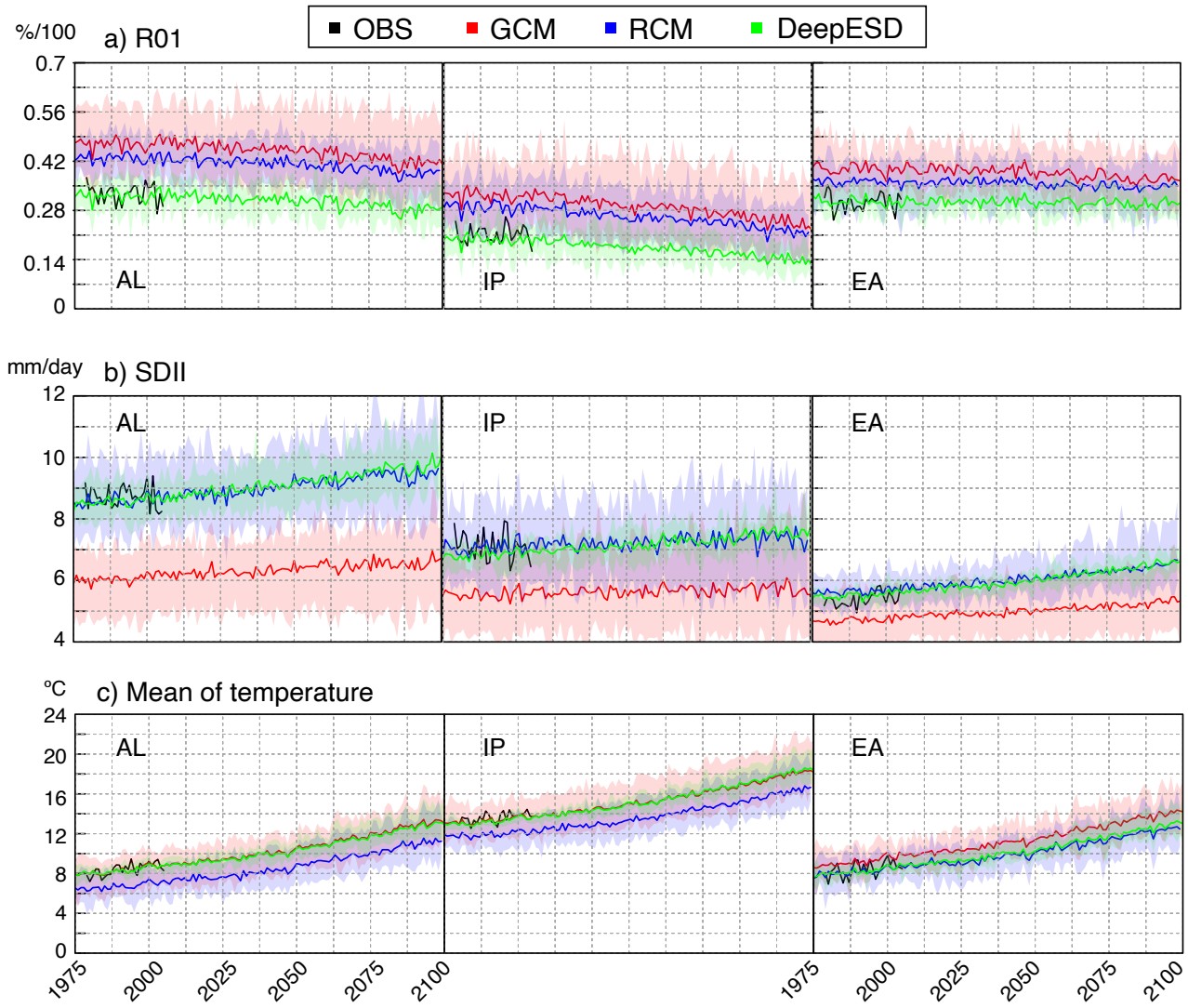

**Figure 6.** Annual time-series for R01, SDII and the mean of temperature, averaged over the eight PRUDENCE regions. For every indicator, the ensemble of GCMs (red), RCMs (blue) and DeepESD (yellow) for the total period 1975-2100, plus the observational reference, E-OBS (black), for the period 1979-2008, are shown. In all cases, the solid lines represent the multi-model ensemble mean whilst the shadows encompass all the models contributing to the ensemble.

scenario. Note that the GCMs time-series are mostly unbiased which is the result of averaging out the positive and negative biases appearing in the spatial fields of Figure 2. As per the projected signals of change, the three ensembles point out to a (quasi) linear increase along the century and across all regions, with warming values of about 4-6°C for the far-future in most of cases.

This indicates that DeepESD is able to accurately reproduce the historical climate —even the discrete-continuous nature of precipitation,— and beyond the regional differences showed in Figure 5, there is a synchrony in the temporal evolution of the signals among ensembles. These results also indicate that DeepESD results in a smaller spread of the ensemble due to the adjustment of the models towards the observed climatology.

## 4   Conclusions

Deep learning topologies are increasingly being tested for downscaling purposes, achieving promising results in present climate due to their ability to infer complex non-linear patterns from climate data. Nevertheless, the ability of these models to generalize to out-of-sample climate change conditions is still to be analyzed with many questions open. Here, we present DeepESD, an ensemble of regional precipitation and temperature projections (up to 2100) over Europe produced by applying convolutional neural networks to downscale a set of eight GCMs over the EUR-44 CORDEX domain. This multi-model perspective permits to analyze unexplored aspects of CNN-based downscaling such as the inter-model uncertainty of the climate change signals or the similarities/differences of the downscaling across GCMs. We build on existing CNNs models (Baño-Medina et al., 2020) and focus on their performance in the climate model space, using GCM projections. In this sense, we follow previous literature (Vrac et al., 2007; San-Martín et al., 2017; Quesada-Chacón et al., 2021) and compare the DeepESD future fields with a set of state-of-the-art CORDEX RCMs, which are used as "pseudo-observations". To our knowledge, this is the first time that CNNs have been used to produce downscaled multi-model ensembles based on the perfect-prognosis approach and are compared against an ensemble of RCMs.

We find that CNN-based downscaling is able to reproduce the observed climate over the historical period for both precipitation and temperature fields at a distributional level, reducing the systematic biases exhibited by the global and regional physical models. When analyzing the future climate change signals we find that DeepESD presents spatial patterns and magnitudes which are broadly similar to the ones from the RCMs. Nevertheless, there are regional differences —at a climatological, and inter-annual scales,— in the projected climate change signals among DeepESD and the physical-based models. For the case of precipitation, these differences are driven towards a decrease of the multi-model uncertainty with respect to the one of their driving GCMs. As per temperature, the CNNs project similar signals of change than the GCMs being able to capture the particularities of each one resulting into a similar ensemble spread. This property was not perceived in previous studies (Baño-Medina et al., 2021) where a single GCM (i.e., EC-Earth) was considered.

Despite the analysis presented herein, the plausibility of the projections has to be further analyzed prior to the integration of DL topologies into climate change applications. For instance, this can be done by developing specific studies dealing with the domain adaptation of the statistical models learned in "perfect" conditions to climate model spaces; by conducting synthetic case-studies permitting to analyze their extrapolation capabilities to climate change conditions; and by comparing the CNN-based fields against other machine learning techniques. To this aim, following the FAIR principles we make publicly available DeepESD from the ESGF portal, which will allow the scientific community to continue exploring the benefits and shortcomings of these new techniques for the downscaling of climate. Precisely, DeepESD contributes to CORDEX EUR-44 being the first

statistical-based dataset to ever participate in this international initiative, entailing a breakthrough of this type of techniques on the study of regional climate.

*Code and data availability.* To promote transparency and reproducibility of our results, we provide the data (DOI: 10.5281/zenodo.6823421) and the companion Jupyter notebook (DOI: 10.5281/zenodo.6828303), explaining how DeepESD has been produced. This notebook is based on the `R` software and builds on the `climate4R` framework, a set of libraries specifically designed for climate data access and post-processing (Iturbide et al., 2019). To build the CNNs used, we rely on `downscaleR.keras` ((Baño-Medina et al., 2020)), which integrates `Keras`, a state-of-the-art DL library, within `climate4R`. Furthermore, most of the results shown in this manuscript can be replicated by following the indications given in the notebook, providing thus the basis for practitioners to perform their own experiments.

DeepESD downscaled results have been published at the ESGF data node at Cantabria University https://data.meteo.unican.es/thredds/catalog/esgcet/catalog.html.

*Author contributions.* J.B., R.M. and J.M.G. conceived the experiment, J.B. produced all the results, E.C. and A.S.C. prepared the data for publication. All authors contributed to the analysis of results and to the writing of the manuscript.

*Competing interests.* The authors declare no competing interests.

*Acknowledgements.* The authors would like to acknowledge the E-OBS dataset from the EU-FP6 project UERRA (http://www.uerra.eu) and the Copernicus Climate Change Service, and the data providers in the ECA&D project (https://www.ecad.eu). We also acknowledge the support from the Spanish Government through project PID2020-116595RB-I00: Contribución a la nueva generación de proyecciones climáticas regionales de CORDEX mediante técnicas dinámicas y estadísticas (CORDyS) funded by MCIN/AEI /10.13039/501100011033. Also, J.B-M acknowledges support from Universidad de Cantabria and Consejería de Universidades, Igualdad, Cultura y Deporte del Gobierno de Cantabria via the "instrumentación y ciencia de datos para sondear la naturaleza del universo" project.

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
