# Peer review of "Downscaling Multi-Model Climate Projection Ensembles with Deep Learning (DeepESD): Contribution to CORDEX EUR-44"

_Geoscientific Model Development, 2022_

## Referee Comment (RC1)

**General comments:**

In this paper, the author built a statistical downscaling model based on (Convolutional Neural Network) CNN. And used the reanalysis data as a reference for the training phase. Then, the author employed the trained model to downscale the ensemble of CMIP5 models over Europe. The results were compared with the CORDEX RCMs for the historical (1975-2005) and the projection (2006-2100) period. The results from the deepSD algorithm are contributing to CORDEX initiative, which is a breakthrough since the statistical downscaling based on artificial intelligence was not trusted for climate studies in the past few years. In general terms, the manuscript is well structured, the methodology is well described, the figures and tables are well organized and the results are adequately discussed. This paper falls within the scope of this journal. In this sense, the article can be approved after minor revisions.

*Recommendation:* Accept after minor revisions.

**Specific comments:**

**Abstract:**

The sentence "To our knowledge, this is the first time that CNNs have been used to produce multi-model ensembles" is not that accurate since there are previous studies that employed CNN to downscale the model ensemble (e.g., Babaousmail et al. (2021)).

**Introduction:**

1. Introduction (third paragraph): "These methods are not computationally demanding…" This sentence needs a citation.
2. The author should justify why he selected the RCP8.5 scenario out of the other scenarios? Also, was there any method employed for the selection of the 8 GCMs?
3. The author didn't justify why E-OBS v20 was selected as an observation in this study.

**Data and methods:**

1. Since the author is comparing the ensemble resulting from 8 GCMs with the RCM ensemble projection, shouldn't be the number of GCMs equal to the number of RCMs?

2. Usually, when we train a neural net model, a validation phase is required after the training and it should be selected from the historical 25 years period, in this paper the author didn't mention it.

3. Concerning the CNN algorithm, we noticed that the CNN used to downscale precipitation has one more layer than the one for temperature (one output layer). Can the author explain the reason?

---

## Referee Comment (RC2)

The manuscript entitled "Downscaling Multi-Model Climate Projection Ensembles with Deep Learning (DeepESD): Contribution to CORDEX EUR-44" applied a Convolutional Neural Networks (CNN) to downscale present-day and future GCM simulations, with a focus on precipitation and temperature simulation. The paper falls within the scope of the journal. The manuscript is well structured, the method is generally clearly presented, and the results fully support its conclusion. I have a few specific comments regarding the method and analyses.

One common problem I found in the manuscript is the use of specific terms that might be well known in the Deep Learning area but is unfamiliar to me, a GCM and RCM modeler, and people like me. I suppose the authors may want to broaden their impact not only in the DP but also on people working on dynamic modeling. Below I have listed a few:

Line 32: "perfect prognosis" please explain the term in detail

Lines 35-36: please include 1-2 sentences to introduce dynamic downscale as a comparison to statistical downscale

Line 61: Please provides more information about "E-OBS v20"

Line 62: I assume "—" is a typo?

Line 70: can you explain the "harmonization process" further?

Lines 75-77: The method of DL should be explained in further detail. "They consist of … (one per each gridpoint in E-OBS)". I found many terms in these sentences that might be barriers to fully understand the method. Can you rephrase it?

Line 97-98: Merge the single sentence to the following paragraph

Line 120: It is interesting that the DeepESD has the smallest ensemble spread over the historical period (Fig. 2) but has the largest one over the future. Any explanation for that?

125: what do "these differences" refer to?

Line 130: "These differences are quite systematic for the case of precipitation indicating a robust CNN extrapolation fingerprint." Can you explain in further detail?

Figure 4: How do you produce Row 3? Is it the difference (DeepESD minus E-OBS v2.0) shown in each model in row 2 minus the mean of the difference? In that case, the mean of 8 panels in row 3 should be 0 ($\overline{Diff} - \overline{\overline{Diff}} = 0$), but clearly they are not.

---

## Referee Comment (RC3)

**Review of "Downscaling Multi-Model Climate Projection Ensembles with Deep Learning (DeepESD): Contribution to CORDEX EUR-44" by Baño-Medina et al., submitted to *Geoscientific Model Development***

**Recommendation:** Major revision

**General comments:**

Recently, the rapid development of deep learning (DL) technologies has provided novel approaches to construct empirical-statistical models for downscaling GCM outputs. DL can identify and extract complex nonlinear relationships that exist in large data sets and model them to develop the downscaling technique. There are progressively more studies exploring a proof of the concept for developing empirical-statistical models based on the DL technique, as in this work.

In this study, the deep learning algorithm of convolutional neural networks (CNN) is used as a statistical downscaling technique. The CNN-based model was trained using daily predictors from the ERA-Interim reanalysis and predictands from E-OBS v20 observations. And then, the well-trained model was applied to downscale eight GCMs outputs for the historical and RCP8.5 periods. The downscaled projections by the CNN-based model were compared with results from RCMs driven by the same GCMs. The authors found that the CNN-based model can provide comparable climate changes signals to those obtained with the RCMs, with a smaller uncertainty for precipitation.

As RCMs require massive computational resources, the CNN-based model may be plausible to be used as an alternative for downscaling GCM outputs. So, please add some discussions about the computational efficiency of the CNN-based model.

This study shows promising results and the manuscript is well written. However, before being accepted, I think the manuscript could be improved as commented below. I believe my comments/suggestions are not very critical and do not require hard work, and this manuscript would be suitable for publication after appropriate revision.

**Specific comments:**

Line 12: "… but a similar uncertainty for temperature"
    More precisely, it is a larger uncertainty for temperature, as shown in Figure 3.

Line 71−73: The CNN-based model provides smaller uncertainties for precipitation. Does the harmonization process contribute to this respect? Please add discussions about the role of the harmonization process.

Figure 1. Please add units (°C) to the color bar of the temperature bias.

Line 102. The CNN-based model is trained with the observations. So it can be expected that DeepESD exhibits a largely unbiased spatial pattern. The CNN-based model is also used to downscale the GCM outputs for the historical period. How about the downscaled historical GCM simulations by the CNN-based model? Please add discussions about the bias of the CNN-based model for the historical period.

L106. Figure 2 shows that DeepESD shows good performance in reproducing the variability and extremes. I think this capability is important. Please add more discussions about Figure 2.

L114−116: I do not quite understand this sentence. Could you please explain it?

---

## Author Comment (AC1)

We thank the reviewer for his/her fruitful comments and sincerely acknowledge his/her time for reviewing the manuscript.

- The sentence "To our knowledge, this is the first time that CNNs have been used to produce multi-model ensembles" is not that accurate since there are previous studies that employed CNN to downscale the model ensemble (e.g., Babaousmail et al. (2021)).

  We thank the reviewer for this interesting reference. We have noticed that in [1], the Deep Learning topology is used as a Model Output Statistics (MOS) technique, since it uses the precipitation —which is also the target variable— of the Generalized Circulation Model (GCM) as input data. This differs with our approach which falls in the "perfect-prognosis" family within statistical downscaling, using reanalysis data as input to calibrate the model. For this reason we have rephrased the sentence indicated by the reviewer to the following one: "To our knowledge, this is the first time that CNNs have been used to produce downscaled multi-model ensembles based on the perfect-prognosis approach".

  .

  [1] Babaousmail, Hassen, et al. "Novel statistical downscaling emulator for precipitation projections using deep Convolutional Autoencoder over Northern Africa." *Journal of Atmospheric and Solar-Terrestrial Physics* 218 (2021): 105614.

- Introduction (third paragraph): "These methods are not computationally demanding…". This sentence needs a citation.

  As compared with the Regional Climate Models (RCMs), which require computational timings from month to years in their simulations, statistical models including convolutional neural networks can be calibrated from hours to days. We have included the following citations:
  *[1] Le Roux, Renan, et al. "Comparison of statistical and dynamical downscaling results from the WRF model." Environmental modelling & software 100 (2018): 67-73.*
  *[2] Baño-Medina, Jorge, Rodrigo Manzanas, and José Manuel Gutiérrez. "Configuration and intercomparison of deep learning neural models for statistical downscaling." Geoscientific Model Development 13.4 (2020): 2109-2124.*

- The author should justify why he selected the RCP8.5 scenario out of the other scenarios?
  We have included the following sentence in the manuscript "We follow previous work in the field [1,2] and select the RCP8.5 scenario which shows the strongest climate change signal (especially for temperature) and, therefore, permits to optimally explore the extrapolation capability of the CNNs."

[1] *ME, Olmo, Rocio Balmaceda-Huarte, and Maria Laura Bettolli. "Multi-model ensemble of statistically downscaled GCMs over southeastern South America: historical evaluation and future projections of daily precipitation with focus on extremes." Climate Dynamics (2022): 1-18.*
[2] *Baño-Medina, Jorge, Rodrigo Manzanas, and José Manuel Gutiérrez. "On the suitability of deep convolutional neural networks for continental-wide downscaling of climate change projections." Climate Dynamics 57.11 (2021): 2941-2951.*

- Also, was there any method employed for the selection of the 8 GCMs?
  The main reason for selecting this ensemble is because it has been already used in other studies so it allows comparison. Moreover, predictors are publicly available (allowing reproducibility, as illustrated in the companion notebook) and have been assessed in a previous study [1]..[1] *Brands, Swen, et al. "How well do CMIP5 Earth System Models simulate present climate conditions in Europe and Africa?." Climate dynamics 41.3 (2013): 803-817.*

- The author didn't justify why E-OBS v20 was selected as an observation in this study.
  We have added the following paragraph to the manuscript:
  "E-OBS is a high-resolution observational dataset generated through an interpolation procedure of the European Climate Assessment & Dataset (ECA&D, [1]) station network. Whilst national and sub-national datasets exist, E-OBS accurately represents the regional climate over the entire European continent [2] and it is commonly employed in statistical downscaling experiments at a continental level [3,4,5,6]. We chose version 20 (v20, release date October 2019) since it was the latest one at the beginning of this study."

[1] Klok, E. J., and A. M. G. Klein Tank. "Updated and extended European dataset of daily climate observations." *International Journal of Climatology: A Journal of the Royal Meteorological Society* 29.8 (2009): 1182-1191.
[2] *Bandhauer, Moritz, et al. "Evaluation of daily precipitation analyses in E‐OBS (v19. 0e) and ERA5 by comparison to regional high‐resolution datasets in European regions." International Journal of Climatology 42.2 (2022): 727-747.*
[3] *Maraun, Douglas, et al. "VALUE: A framework to validate downscaling approaches for climate change studies." Earth's Future 3.1 (2015): 1-14.*
[4] *Vrac, Mathieu, and Pradeebane Vaittinada Ayar. "Influence of bias correcting predictors on statistical downscaling models." Journal of Applied Meteorology and Climatology 56.1 (2017): 5-26.*
[5] *Baño-Medina, Jorge, Rodrigo Manzanas, and José Manuel Gutiérrez. "Configuration and intercomparison of deep learning neural models for statistical downscaling." Geoscientific Model Development 13.4 (2020): 2109-2124.*
[6] *Baño-Medina, Jorge, Rodrigo Manzanas, and José Manuel Gutiérrez. "On the suitability of deep convolutional neural networks for continental-wide downscaling of climate change projections." Climate Dynamics 57.11 (2021): 2941-2951.*

- Since the author is comparing the ensemble resulting from 8 GCMs with the RCM ensemble projection, shouldn't be the number of GCMs equal to the number of RCMs?
  As the reviewer mentions, ideally we would have the same number of members in both GCM and RCM ensembles. Under this assumption there would only be 1 RCM

per GCM resulting in a total of 8 members for both ensembles. However, we wanted to avoid possible artifacts in the results due to a lack of variability in the RCM selection. For this reason we occasionally utilize 2 RCMs per GCM, representing a compromise between having a similar —but not equal— number of members for each ensemble and partially including this source of uncertainty in the results.

- Usually, when we train a neural net model, a validation phase is required after the training and it should be selected from the historical 25 years period, in this paper the author didn't mention it.
  During training, we use a validation set (10% of the data) to cross-validate the results performing early-stopping [1] (the training stops when the validation test start increasing). We have included the following phrase in the annex of the manuscript to mention this aspect: "During calibration, we use a validation set (10% of the data randomly selected) to perform early-stopping [1], and finish the training whenever the validation loss stops decreasing after 30 epochs" . After training, the validation in ''perfect'' conditions was already carried out in [1] and it is appropriately cited in lines 40 and 74-75.

  [1] *Baño-Medina, Jorge, Rodrigo Manzanas, and José Manuel Gutiérrez. "Configuration and intercomparison of deep learning neural models for statistical downscaling." Geoscientific Model Development 13.4 (2020): 2109-2124.*

- Concerning the CNN algorithm, we noticed that the CNN used to downscale precipitation has one more layer than the one for temperature (one output layer). Can the author explain the reason?
  The configuration of the DeepESD method was undertaken in [1] and here we used the optimum configurations found in that study. To downscale both temperature and precipitation fields, the CNNs deployed contain the same number of hidden layers (3) but different number of, n, output neurons ($2*n$ for temperature and $3*n$ for precipitation; please note that output layers are arranged in parallel and not sequentially), representing each of the statistical parameters of the parametrics distributions estimated per gridbox. For temperature, we learn Gaussian daily conditional distributions parameterized by 2 parameters (mean and standard deviation) per predictand site, while for precipitation there are 3 parameters (probability of rain, shape and scale factor) corresponding to the Bernoulli-Gamma distribution. This results in 2 and 3 output layers in the convolutional topology, respectively. More details can be found in the reference describing the convolutional network here employed [1], already cited in the manuscript.

  [1] *Baño-Medina, Jorge, Rodrigo Manzanas, and José Manuel Gutiérrez. "Configuration and intercomparison of deep learning neural models for statistical downscaling." Geoscientific Model Development 13.4 (2020): 2109-2124.*

---

## Author Comment (AC2)

We thank the reviewer for his/her fruitful comments and sincerely acknowledge his/her time for reviewing the manuscript.

- Line 32: "perfect prognosis" please explain the term in detail
  We have extended the explanation of "perfect-prognosis" resulting into the following paragraph:
  "Under the "perfect-prognosis" (PP) approach, the statistical models are trained using observations (both for predictors and predictands) in a historical period and afterwards applied to GCM predictors from global projections to obtain the regional/local downscaled results. In particular, reanalysis data are used for the predictor set during calibration. Several assumptions need to be fulfilled in any PP downscaling setup [1]; (a) predictors have to be realistically simulated by GCMs and project with certain plausibility the climate change signal, (b) these predictors to be informative to the regional scale, and (c) the statistical models have to be flexible enough to learn the complex patterns inherent in the predictor-predictand link."

  [1] Gutiérrez, José Manuel, et al. "An intercomparison of a large ensemble of statistical downscaling methods over Europe: Results from the VALUE perfect predictor cross‐validation experiment." International journal of climatology 39.9 (2019): 3750-3785.

- Lines 35-36: please include 1-2 sentences to introduce dynamic downscale as a comparison to statistical downscale
  We have included the following sentence: "As compared to DD, PP lacks explicit physics in the model formulation, but overcomes systematic biases present in RCM products, since the model is trained using observations. Regarding computational requirements PP has smaller requirements avoiding the need for large computational infrastructures [1,2].
  [1] Le Roux, Renan, et al. "Comparison of statistical and dynamical downscaling results from the WRF model." Environmental modelling & software 100 (2018): 67-73.
  [2] Baño-Medina, Jorge, Rodrigo Manzanas, and José Manuel Gutiérrez. "Configuration and intercomparison of deep learning neural models for statistical downscaling." Geoscientific Model Development 13.4 (2020): 2109-2124.

- Line 61: Please provides more information about "E-OBS v20"
  We have added the following paragraph to the manuscript:
  "E-OBS is a high-resolution observational dataset generated through an interpolation procedure of the European Climate Assessment & Dataset (ECA&D, [1]) station network. Whilst national and sub-national datasets exist, E-OBS accurately represents the regional climate over the entire European continent [2] and it is commonly employed in statistical downscaling experiments on a continental level [3,4,5,6]. We chose version 20 (v20, release date October 2019) since it was the latest one at the beginning of this study."

  [1] Klok, E. J., and A. M. G. Klein Tank. "Updated and extended European dataset of daily climate observations." International Journal of Climatology: A Journal of the Royal Meteorological Society 29.8 (2009): 1182-1191.
  [2] Bandhauer, Moritz, et al. "Evaluation of daily precipitation analyses in E‐OBS (v19. 0e) and ERA5 by comparison to regional high‐resolution datasets in European regions." International Journal of Climatology 42.2 (2022): 727-747.

[3] *Maraun, Douglas, et al. "VALUE: A framework to validate downscaling approaches for climate change studies." Earth's Future 3.1 (2015): 1-14.*
[4] *Vrac, Mathieu, and Pradeebane Vaittinada Ayar. "Influence of bias correcting predictors on statistical downscaling models." Journal of Applied Meteorology and Climatology 56.1 (2017): 5-26.*
[5] *Baño-Medina, Jorge, Rodrigo Manzanas, and José Manuel Gutiérrez. "Configuration and intercomparison of deep learning neural models for statistical downscaling." Geoscientific Model Development 13.4 (2020): 2109-2124.*
[6] *Baño-Medina, Jorge, Rodrigo Manzanas, and José Manuel Gutiérrez. "On the suitability of deep convolutional neural networks for continental-wide downscaling of climate change projections." Climate Dynamics 57.11 (2021): 2941-2951.*

- Line 62: I assume "—" is a typo?
  Yes, thank you for noticing it.

- Line 70: can you explain the "harmonization process" further?
  We have included the following paragraph:
  "Moreover, we have applied a harmonization process suggested in previous works [1,2,3], to increase the distributional similarity between the GCM and reanalysis fields. This post-processing consists of bias adjusting the seasonal cycle of the historical and RCP8.5 predictor variables towards the one described by the reanalysis fields over the reference period 1979-2005. A simple adjustment (mean and variance) applied at a monthly scale is used to keep this harmonization as simple as possible.

  [1] *Brands, Swen, et al. "Validation of the ENSEMBLES global climate models over southwestern Europe using probability density functions, from a downscaling perspective." Climate Research 48.2-3 (2011): 145-161.*
  [2] *Vrac, Mathieu, and Pradeebane Vaittinada Ayar. "Influence of bias correcting predictors on statistical downscaling models." Journal of Applied Meteorology and Climatology 56.1 (2017): 5-26.*
  [3] *Baño-Medina, Jorge, Rodrigo Manzanas, and José Manuel Gutiérrez. "On the suitability of deep convolutional neural networks for continental-wide downscaling of climate change projections." Climate Dynamics 57.11 (2021): 2941-2951.*

- Lines 75-77: The method of DL should be explained in further detail. "They consist of … (one per each gridpoint in E-OBS)". I found many terms in these sentences that might be barriers to fully understand the method. Can you rephrase it?
  We agree with the reviewer that some concepts related to the deep learning terminology (e.g., convolutional layer, filter map and kernel), which are unfamiliar for non-machine-learning researchers, were hardly explained in the manuscript, difficulting the understanding of the proposed topology. For this reason we have added the following sentences:
  **(introduction)** A 2-D convolutional layer convolutes a set of k = 1, 2, ..., K 2-D parameters called kernel over the input space (or previous hidden layer), generating K filter maps, which are spatial representations of the K patterns optimized by the network (we refer the reader to [1] for more details)
  **(Data and Methods)** In particular, we deploy the best performing topologies developed in [1], a recent study which intercompares different CNNs over Europe in ``perfect" conditions  to downscale precipitation/temperature. They consist of 3-layers

with 3x3 kernels of 50, 25 and 1/10 filter maps followed by a dense connection which links all the neurons in the last hidden layer to the output neurons (one per each land gridpoint in E-OBS). For precipitation (temperature), these CNNs are trained to optimize the negative log-likelihood of a Bernoulli-Gamma (Gaussian) distribution, yielding thus daily estimates of 3 (2) parameters per predictand site (e.g., for precipitation $n^o_{output\ neurons} = n^o_{predictand\ sites} * 3$) representing the probability of rain, shape and scale (mean and variance). We refer the reader to [1] for a detailed computational analysis and more details in the topology.

[1] *Baño-Medina, Jorge, Rodrigo Manzanas, and José Manuel Gutiérrez. "Configuration and intercomparison of deep learning neural models for statistical downscaling." Geoscientific Model Development 13.4 (2020): 2109-2124.*

- Line 97-98: Merge the single sentence to the following paragraph
  Done.

- Line 120: It is interesting that the DeepESD has the smallest ensemble spread over the historical period (Fig. 2) but has the largest one over the future. Any explanation for that?
  In the historical period the predictor fields are all standardized with the same mean and standard deviation —regardless of the GCM considered,— thus leading to a small ensemble spread. Nevertheless, in the future the ensemble spread of DeepESD is the result of 1) the divergence in the trends/evolution of the predictor fields, and/or to 2) the extrapolation ability of the CNN.

- 125: what do "these differences" refer to?
  It refers to the differences in the climate change signal between the GCM and the RCM/DeepESD, which are described in the preceding paragraph. In particular, the differences between the GCM and the DeepESD are displayed in rows 1-2 of Figure 4.

- Line 130: "These differences are quite systematic for the case of precipitation indicating a robust CNN extrapolation fingerprint." Can you explain in further detail?
  Thank you for this comment. We agree that this sentence may be misleading. We have decided to replace it by the following one: "For the case of precipitation, DeepESD modifies the signal differently for each GCM, pushing the values closer across GCMs and thus reducing the inter-model uncertainty (Figure 3)."

- Figure 4: How do you produce Row 3? Is it the difference (DeepESD minus E-OBS v2.0) shown in each model in row 2 minus the mean of the difference? In that case, the mean of 8 panels in row 3 should be 0, but clearly they are not.
  In row 2 we display the difference (Diff) between the climate change signal for mean temperature projected by DeepESD (Delta$_{CNN}$) and the one projected by the GCM (Delta$_{GCM}$). In row 3, for each GCM (i.e., column) we compute the difference between Diff and the spatial average of Diff ($\overline{Diff}$). Since Diff is a spatial field (row 2), whilst $\overline{Diff}$ is just a number, the operation Diff - $\overline{Diff}$ can be different from 0.

---

## Author Comment (AC3)

We thank the reviewer for his/her fruitful comments and sincerely acknowledge his/her time for reviewing the manuscript.

- As RCMs require massive computational resources, the CNN-based model may be plausible to be used as an alternative for downscaling GCM outputs. So, please add some discussions about the computational efficiency of the CNN-based model.
  We have included the following sentence: "As compared to DD, PP lacks explicit physics in the model formulation, but overcomes systematic biases present in RCM products, since the model is trained using observations. Regarding computational requirements PP has smaller requirements avoiding the need for large computational infrastructures [1,2]. These aspects make PP models attractive to be extensively used to downscale global multi-model ensembles providing continental-wide regional projection fields, e.g. over the CORDEX domains, a key task which is mostly undertaken by means of DD nowadays"

  [1] Le Roux, Renan, et al. "Comparison of statistical and dynamical downscaling results from the WRF model." Environmental modelling & software 100 (2018): 67-73.
  [2] Baño-Medina, Jorge, Rodrigo Manzanas, and José Manuel Gutiérrez. "Configuration and intercomparison of deep learning neural models for statistical downscaling." Geoscientific Model Development 13.4 (2020): 2109-2124.

  We have also added the computational resources and timings required for training and prediction of our particular use-case in the manuscript: "We lean on a 2x Intel(R) Xeon(R) CPU E5-2670 0 @ 2.60GHz (16 cores, 32 threads) with 60 GB of memory RAM. The computational time employed to calibrate the model and generate the projections for e.g., one GCM is approximately 5-6 hours, being considerably lower than the one needed to run a RCM (weeks to months)"
  Moreover, we cite [2] which compares the computational efficiency of the same CNN topology here deployed against other statistical models.

  [1] Le Roux, Renan, et al. "Comparison of statistical and dynamical downscaling results from the WRF model." Environmental modelling & software 100 (2018): 67-73.
  [2] Baño-Medina, Jorge, Rodrigo Manzanas, and José Manuel Gutiérrez. "Configuration and intercomparison of deep learning neural models for statistical downscaling." Geoscientific Model Development 13.4 (2020): 2109-2124.

- Line 12: "… but a similar uncertainty for temperature". More precisely, it is a larger uncertainty for temperature, as shown in Figure 3.
  Yes, thank you for noticing it. We have rephrased it to "... but a larger uncertainty for temperature".

- Line 71−73: The CNN-based model provides smaller uncertainties for precipitation. Does the harmonization process contribute to this respect? Please add discussions about the role of the harmonization process.
  In the historical period the predictor fields are all harmonized (bias adjusting the mean and variance of the seasonal cycle) with ERA-Interim —regardless of the GCM considered,— thus leading to a small ensemble spread. Nevertheless, in the future the ensemble spread of DeepESD is the result of 1) the divergence in the

trends/evolution of the predictor fields, and/or to 2) the extrapolation ability of the CNN. Nevertheless, we agree with the reviewer that the harmonization may alter the trends in the projected signals [1] and may have an impact on the inter-model uncertainty. We are currently exploring the influence of different bias adjustment techniques over the predictor space in the downscaled signal, and therefore a more extensive analysis on this task will be published as future work. In the meantime we have added these two sentences in the manuscript:

**(Data and Methods)** "Since advanced bias adjustment techniques may alter the raw climate change signal [1], we only adjust the mean and variance at a monthly scale to keep this as simple as possible."

**(conclusions)** "Also, the sensitivity of the projected signals to the bias adjustment of the predictor fields is to be explored in future work."

[1] *Casanueva, Ana, et al. "Testing bias adjustment methods for regional climate change applications under observational uncertainty and resolution mismatch." Atmospheric Science Letters 21.7 (2020): e978.*

- Figure 1. Please add units (°C) to the color bar of the temperature bias.
  Done. Thank you for noticing it.

- Line 102. The CNN-based model is trained with the observations. So it can be expected that DeepESD exhibits a largely unbiased spatial pattern. The CNN-based model is also used to downscale the GCM outputs for the historical period. How about the downscaled historical GCM simulations by the CNN-based model? Please add discussions about the bias of the CNN-based model for the historical period.
  Indeed, the bias pattern displayed in Figure 1, row 2, columns 3 and 6, is the climatological difference of the downscaled historical mean of precipitation and temperature obtained with DeepESD, and the same field for the raw GCM. So this figure is already the one demanded by the reviewer and is discussed in the manuscript. We have clarified this aspect by adding the following sentence: " The bottom row displays the corresponding biases of the raw (GCM) and downscaled historical simulations (RCM and DeepESD) with respect to E-OBS v20."

  Moreover, as shown in [1], the bias of the downscaled mean of temperature and precipitation for the same reference period using ERA-Interim predictors as input data, shows also a mostly unbiased pattern.

  [1] *Baño-Medina, Jorge, Rodrigo Manzanas, and José Manuel Gutiérrez. "Configuration and intercomparison of deep learning neural models for statistical downscaling." Geoscientific Model Development 13.4 (2020): 2109-2124.*

- L106. Figure 2 shows that DeepESD shows good performance in reproducing the variability and extremes. I think this capability is important. Please add more discussions about Figure 2.
  We have added the following paragraph in the manuscript:
  "Besides these results for the mean, Figure 2 compares the entire precipitation (>= 1 mm/day) and temperature distributions for the GCM (red), RCM (blue) and DeepESD

(green) ensembles over the historical period 1979-2005, plus E-OBS (black), for the Alps, Iberian Peninsula and Eastern Europe as illustrative for different European climates. The solid line represents the ensemble mean and the shadow encompasses two standard deviations. The dashed line indicates the distributional mean of each PDF. For precipitation, we observe a good fit between E-OBS and DeepESD. This is a direct consequence of sampling from the inferred conditional Gamma distributions. RCMs and, particularly, GCMs overestimate low rainfall events and underestimate the high rainfall ones. Although the differences among the three ensembles for temperature are not as notable as for precipitation, DeepESD tends to follow more accurately the E-OBS curve than the GCM and the RCM."

[1] *Carreau, Julie, and Mathieu Vrac. "Stochastic downscaling of precipitation with neural network conditional mixture models." Water Resources Research 47.10 (2011).*

- L114−116: I do not quite understand this sentence. Could you please explain it?

  We agree with the reviewer that the sentence might be confusing. We have rephrased it to: "However, slight regional differences exist among the three ensembles, especially when compared with GCMs, with DeepESD presenting weaker signals of change over the Iberian Peninsula and more intense signals over some parts of Northern and Eastern Europe."

---

## Author Comment (AC4)

We thank the reviewer for his/her fruitful comments and sincerely acknowledge his/her time for reviewing the manuscript.

- With which criteria were chosen the predictand fields?
  We chose to downscale air surface temperature and daily total precipitation since they are two variables highly demanded by the climate, impact and adaptation communities. These are also the variables with best observations available and, therefore, provide reliable training data sets. It would be interesting to extend this study to other relevant predictand variables (e.g., wind) and we leave this possible continuation to future work.

- How expensive in computer resources is the method?
  We have included the following sentence in the manuscript:
  "We lean on a 2x Intel(R) Xeon(R) CPU E5-2670 0 @ 2.60GHz (16 cores, 32 threads) with 60 GiB of memory RAM. The computational time employed to calibrate the model and generate the projections for e.g., one GCM is approximately 5-6 hours, being considerably lower than the one needed to run a RCM."

- Why eobs was used? It is too smooth, what can be seen in the results, specially in places with high topography. Why did not used a regional, high resolution reanalysis as predictands?
  We have added the following paragraph to the manuscript:
  "E-OBS is a high-resolution observational dataset generated through an interpolation procedure of the European Climate Assessment & Dataset (ECA&D, [1]) station network. Whilst national and sub-national datasets exist, E-OBS accurately represents the regional climate over the entire European continent [2] and it is commonly employed in statistical downscaling experiments on a continental level [3,4,5,6]. We chose version 20 (v20, release date October 2019) since it was the latest one at the beginning of this study."

  [1] Klok, E. J., and A. M. G. Klein Tank. "Updated and extended European dataset of daily climate observations." *International Journal of Climatology: A Journal of the Royal Meteorological Society* 29.8 (2009): 1182-1191.
  [2] *Bandhauer, Moritz, et al. "Evaluation of daily precipitation analyses in E‐OBS (v19. 0e) and ERA5 by comparison to regional high‐resolution datasets in European regions." International Journal of Climatology 42.2 (2022): 727-747.*
  [3] *Maraun, Douglas, et al. "VALUE: A framework to validate downscaling approaches for climate change studies." Earth's Future 3.1 (2015): 1-14.*
  [4] *Vrac, Mathieu, and Pradeebane Vaittinada Ayar. "Influence of bias correcting predictors on statistical downscaling models." Journal of Applied Meteorology and Climatology 56.1 (2017): 5-26.*
  [5] *Baño-Medina, Jorge, Rodrigo Manzanas, and José Manuel Gutiérrez. "Configuration and intercomparison of deep learning neural models for statistical downscaling." Geoscientific Model Development 13.4 (2020): 2109-2124.*
  [6] *Baño-Medina, Jorge, Rodrigo Manzanas, and José Manuel Gutiérrez. "On the suitability of deep convolutional neural networks for continental-wide downscaling of climate change projections." Climate Dynamics 57.11 (2021): 2941-2951.*

- It seems that the use of more output layers for the precipitation than in temperature makes the biases in the downscaling of precipitation as small as for temperature, but reduces the standard deviation in the downscaling (Figure 3) . I think that this fact is related to the methodology and should be commented on by the authors.

  Both temperature and precipitation topologies have the same number of filter maps and hidden layers in their topologies and, thus, they can achieve the same degree of nonlinearity. The CNNs deployed contain different number of, n, output neurons (2*n for temperature and 3*n for precipitation), representing each of the statistical parameters of the parametric distributions estimated per gridbox. For temperature, we learn Gaussian daily conditional distributions parameterized by 2 parameters (mean and standard deviation) per predictand site, while for precipitation there are 3 parameters (probability of rain, shape and scale factor) corresponding to the Bernoulli-Gamma distribution. Having more output layers does not add non-linearity to the network, and therefore this aspect does not have an influence on the future estimates and indices —either biases or standard deviation. For more details on the topology we refer to [1].

  [1] *Baño-Medina, Jorge, Rodrigo Manzanas, and José Manuel Gutiérrez. "Configuration and intercomparison of deep learning neural models for statistical downscaling." Geoscientific Model Development 13.4 (2020): 2109-2124.*

- Also, the fact that the simulation of R01 in DeepESD is closer to the RCMs that to the GCMs shows the importance of a good simulation of orographic precipitation, while SDII and Mean temperature in DeepESD and GCM are closer, probably reflecting the tuning of the GCMs (which usually is not made in RCMs) and the training with observations in DeepESD. The exception for temperature in ED looks strange for me and would be nice if you explain this behavior.
  This aspect is related to what is explained in lines 126-127: "In the case of the RCMs, some recent studies attribute these differences to the lack of time-varying anthropogenic aerosols in the RCM formulation (Boé et al., 2020; Gutiérrez et al., 2020)". Therefore, there is an on-going analysis by the dynamical downscaling community to analyze the differences mentioned by the reviewer in Eastern and Central Europe of the climate change signal of temperature between the GCMs and RCMs, investigating whether it is due to an added value of dynamical downscaling or to deficiencies in the model formulation of the RCMs.

- Results in figures 3,4 and could be also contributed by the use of stochastic (deterministic) approaches for the precipitation (temperature) specific comments
  The choice for either stochastic or deterministic downscaled fields is mainly relevant to the reproduction of extremes (Figure 2), but they do have a negligible influence on the results of Figure 3 and 4, which display only the mean of temperature and precipitation. This comparison among stochastic and deterministic fields can be encountered in [1].

[1] *Baño-Medina, Jorge, Rodrigo Manzanas, and José Manuel Gutiérrez. "On the suitability of deep convolutional neural networks for continental-wide downscaling of climate change projections." Climate Dynamics 57.11 (2021): 2941-2951.*

- A more detailed description of the methodology for not specialists (most readers, I guess) should be interesting. Can be added as an appendix

  We agree with the reviewer that some concepts related to the deep learning terminology (e.g., convolutional layer, filter map and kernel), which are unfamiliar for non-machine-learning researchers, were hardly explained in the manuscript, difficulting the understanding of the proposed topology. For this reason we have added the following sentences:
  "In particular, we deploy the best performing topologies developed in [1], a recent study which intercompares different CNNs over Europe in ``perfect" conditions to downscale precipitation/temperature. They consist of 3-layers with 3x3 kernels of 50, 25 and 1/10 filter maps followed by a dense connection which links all the neurons in the last hidden layer to the output neurons at each land gridpoint in E-OBS. For precipitation (temperature), these CNNs are trained to optimize the negative log-likelihood of a Bernoulli-Gamma (Gaussian) distribution, yielding thus daily estimates of 3 (2) parameters per predictand site (e.g., for precipitation $n^{o}_{\text{output neurons}}$ = $n^{o}_{\text{predictand sites}}$ * 3) representing the probability of rain, shape and scale (mean and variance). We refer the reader to [1] for a detailed computational analysis and more details in the topology."

- How does the interpolation method influence the results?
  To analyze this issue we used two different approaches to re-grid the GCM predictor fields to a common 2° latitude-longitude: nearest-neighbour and bilinear interpolation. We found no remarkable differences in the climate change signals obtained for these two interpolation methods. We will add a comment on this to the revised manuscript.

- In the Iberian Peninsula and the Scandinavian peninsula the climate change signal in DeepESD is similar to that of the global models, while the opposite is true in central Europe. Could you elaborate on this?
  We agree with the reviewer that this aspect is very interesting. To understand the nature of these similarities and differences between the GCM and DeepESD ensembles —beyond a comparison with RCMs, which is what is done in this manuscript,— we have designed a few experiments that we plan to publish in a future paper (outlined in lines 185-189). These experiments consist on analyzing (1) the adaptability of the statistical model learned in perfect conditions (i.e., with reanalysis data for the predictors) to the climate model space, (2) the extrapolation ability of CNNs on pseudo-reality experiments, and (3) the influence of each predictor in the climate change signal.

---

## Author Comment (AC5)

We thank the reviewer for his/her fruitful comments and sincerely acknowledge his/her time for reviewing the manuscript.

- Could you make flowchart of your workflow and include as a figure in the methods section? It is a bit hard to follow your exact procedure.

We will included an schematic flowchart in the revised manuscript. A draft is included below.

[Figure]

- Isn't the comparison to observations between unconstrained mechanistic models (i.e. GCMs) and CNNs trained on observations "unfair"? If you did some nudging procedure with GCMs you would also end up with model output better fitting observations. For the CNN training, did you split the observational data into train (validation) and test set (only train on 20 years and show performance for 10 years)? Again, a flowchart would help to understand what you did. If you show the performance of DeepESD for the test set and compare that to GCM output, it'd be "more fair", but still, just by design we would expect that the CNN reproduces observations better than GCMs.

The flowchart indicated in the previous question probably sheds light on this one as well. Indeed, the train and test sets differ in the predictor datasets (ERA-Interim for training and GCMs for testing) rather than in the temporal period. As the reviewer points out, the comparison between GCM and DeepESD is still not totally "fair" in the historical period. However, in the manuscript we do not intend to establish an argument in this line, but rather to compare the climate change signals between ensembles whilst showing DeepESD a good reproducibility of the local scale in the historical period.

- You show that the CNN learns the "necessary" dynamics based on predictors of the historical period and extrapolates reasonably well using predictors from GCM output for projections. That is a very interesting point. I wonder if this simple bias correction for GCMs really does the trick, as the models considerably diverge over the climatic time-scales and very model specific regional biases emerge. Can you comment on whether other bias-correcting measures were tested?

Overall, there is certainly a long list of potential further evaluation and testing steps that could be undertaken, but maybe it is enough for this model description paper.

We agree with the reviewer that there are many factors that could have some influence in the downscaled results obtained with our CNNs which may be worth analyzing further. For instance, we are currently exploring the influence that the application of different bias correction techniques over the predictor space may have in the downscaled climate change signal. However, in order to not add more complexity to the present manuscript, we plan to publish the results from this analysis in a future paper..

For the moment we have compared the downscaled projections that are obtained based on two different approaches for bias correcting the GCM predictors —(1) bias correction of the mean at a monthly scale following [1], and (2) bias correction of the mean and standard deviation at a monthly scale (DeepESD),— with no major differences found.

[1] *Baño-Medina, Jorge, Rodrigo Manzanas, and José Manuel Gutiérrez. "On the suitability of deep convolutional neural networks for continental-wide downscaling of climate change projections." Climate Dynamics 57.11 (2021): 2941-2951.*

- L5: What is DeepESD standing for? Please introduce acronym before first usage.

  It stands for Deep learning Empirical Statistical Downscaling (DeepESD). We have introduced this acronym in the manuscript.

- LL33-34: The "perfect prognosis" approach is based on the assumption that GCMs don't have systematic biases with respect to the observations that were used for training, right? Maybe you should include a short sentence here that addresses this aspect.

  We will include a comment on this in the revised paper.

- L55: I recommend to use another more static hosting platform for your code, e.g. Zenedo (https://zenodo.org/).

  The code is already hosted in Zenodo. However in the manuscript we point to both the GitHub repository (https://github.com/SantanderMetGroup/DeepDownscaling) and Zenodo (DOI:10.5281/zenodo.3461087) which might be confusing. Also, based on the comment from the chief editor we have removed all references to GitHub and just stick to Zenodo.

- L60: Why did you use ERA-Interim reanalysis? It is outdated for quite some time now.

This study builds on previous ones which use ERA-Interim data to deploy CNNs over Europe [1,2]. For consistency with these studies and also with reference statistical downscaling experiments in the continent [3,4,5,6] which also build on ERA-Interim data, we decided to use this dataset for the predictors. However, we plan to move to ERA5 to downscale CMIP6 GCMs in future work.

[1] *Baño-Medina, Jorge, Rodrigo Manzanas, and José Manuel Gutiérrez. "Configuration and intercomparison of deep learning neural models for statistical downscaling." Geoscientific Model Development 13.4 (2020): 2109-2124.*
[2] *Baño-Medina, Jorge, Rodrigo Manzanas, and José Manuel Gutiérrez. "On the suitability of deep convolutional neural networks for continental-wide downscaling of climate change projections." Climate Dynamics 57.11 (2021): 2941-2951.*
[3] *Bedia, Joaquín, et al. "Statistical downscaling with the downscaleR package (v3. 1.0): contribution to the VALUE intercomparison experiment." Geoscientific Model Development 13.3 (2020): 1711-1735.*
[4] *Maraun, Douglas, et al. "VALUE: A framework to validate downscaling approaches for climate change studies." Earth's Future 3.1 (2015): 1-14.*
[5] *Maraun, Douglas, Martin Widmann, and José M. Gutiérrez. "Statistical downscaling skill under present climate conditions: A synthesis of the VALUE perfect predictor experiment." International Journal of Climatology 39.9 (2019): 3692-3703.*
[6] *Gutiérrez, José Manuel, et al. "An intercomparison of a large ensemble of statistical downscaling methods over Europe: Results from the VALUE perfect predictor cross‑validation experiment." International journal of climatology 39.9 (2019): 3750-3785.*

- L62: I don't understand your use of dashes (—) in the manuscript. Please check whether the make sense throughout the manuscript.

  This character is going to be checked across the entire notebook as the reviewer suggests.

- LL62-65: What about adding high-resolution orography description as static predictor?

  This is done in other studies (e.g., super-resolution, where high-resolution predictors are used to downscale target variables) but is not a standard approach for perfect-prognosis downscaling.

- L85: Why did you analyze both and can you provide the reason why you settled with the deterministic one?

  Sampling from the conditional distributions permits reproducing the temporal variability of the local time-series. However, since sampling is performed at each gridpoint individually, there is a loss in the spatio-temporal structure of the downscaled fields. Since local temperature is largely explained by the large-scale predictors, there is no need to sample from the inferred conditional distributions. Conversely, local precipitation is not completely explained by the predictor set, and a stochastic downscaled version is needed to recover the temporal variability of the target predictand. These aspects were already analyzed in a prior study [1] and for this reason we do not delve into too much detail in this manuscript.

[1] *Baño-Medina, Jorge, Rodrigo Manzanas, and José Manuel Gutiérrez. "On the suitability of deep convolutional neural networks for continental-wide downscaling of climate change projections." Climate Dynamics 57.11 (2021): 2941-2951.*

L88: Please stick to the tenses (in this paragraph you mix present and past tense), i.e. do not switch between present and past tense when describing your results or methods. I recommend that you always use present tense when talking about your study, i.e. when describing your methods, your results etc., and use past tense when referring to already published studies.

Solved

- L137: "contribute to increasing"

Solved.

- Figure 1: Add unit at lower right colorbar. Also, it'd be useful if you could include letter characters as pointers to subplots, e.g. a,b,c,d. This comment applies for all Figures.

We will consider this suggestion in the revised version of the manuscript.

- Figure 4: Please be more specific about the numbers in the plots. Please provide more detailed information in the caption.

Solved.

- Figure 5: The mid-column misses a time axis. DeepESD is not "yellow" but "green", no?

It is green. This typo has been corrected in the new version of the manuscript.

---

## Author Response (AR1)

**Paper gmd-2022-57: "Downscaling Multi-Model Climate Projection Ensembles with Deep Learning" by Baño-Medina et al.**

**GMD decision after the interactive discussion**

Dear Jorge Baño-Medina,

Thank you very much for your response to the interactive comments on your following preprint under peer review for GMD:

gmd-2022-57
Title: Downscaling Multi-Model Climate Projection Ensembles with Deep Learning (DeepESD): Contribution to CORDEX EUR-44
Author(s): Jorge Baño-Medina et al.
MS type: Model evaluation paper
Iteration: Revised submission

Please prepare and submit a revised version of your manuscript if you are confident that you have satisfactorily addressed all comments and that the revised manuscript will meet the high quality standards of GMD (https://www.geoscientific-model-development.net/peer_review/review_criteria.html). In case of doubt, please ask the handling Topical Editor directly whether they would encourage submission of a revised manuscript or not.

We kindly ask you to log in using your Copernicus Office user ID  to upload the files required for the completion of the review process no later than 13 Jul 2022 at:
https://editor.copernicus.org/GMD/review-file-upload/gmd-2022-57

We kindly ask you to provide a detailed point-by-point response to all referee comments and specify all changes in the revised manuscript. The response to the Referees shall be structured in a clear and easy to follow sequence: (1) comments from Referees, (2) author's response, (3) author's changes in manuscript. In addition, please provide a marked-up manuscript version showing the changes made (using track changes in Word or latexdiff in LaTeX). This version should be combined with your response file so that the Topical Editor can clearly identify what changes have been made.

*Response: Dear Editor, thank you for the time and work devoted to our manuscript. We have performed a substantial revision of the manuscript in response to the reviewers' comments. In particular, we have revised the adjustment procedure used to harmonize the GCM predictors, as suggested by some reviewers, adopting a change preserving method (as suggested in the literature). This resulted in a reduced ensemble spread for temperature (comparable to the original GCM spread) allowing us to explain the main issue reported in the original manuscript.*
*We think the revised manuscript addresses the major changes pointed out by the reviewers during the interactive discussion (we are including below point-by-point responses) and hope the paper is suitable for publication in GMD.*

**Paper gmd-2022-57: "Downscaling Multi-Model Climate Projection Ensembles with Deep Learning" by Baño-Medina et al.**

**Comment from the Chief Editor**

After checking your manuscript, it has come to our attention that it does not comply with our Code and Data Policy.
https://www.geoscientific-model-development.net/policies/code_and_data_policy.html
We can not accept the UDCat the Universidad de Cantabria as the repository for the data. According to our policy, it does not comply with the requirements (funding secured for decades, independence of a single institution, etc.) Moreover, after login into the portal to access the ECMWF_ERA-Interim-ESD data, it is necessary to request approval. This is against our policy; it would be the same that simply stating "data available upon request", and it compromises the replicability of the work. You should move the data necessary to a public repository that complies with our policy (e.g. Zenodo, which you already use). Without being fully familiar with your work, I understand that a complete dataset of ERA fields can have a substantial size (several hundreds of GB) that prevents it from being feasible. Please save a small dataset of predictors as a sample that improves the replicability in such a case. Also, be aware that in this way, the Jupyter notebooks should link the new repositories and not the servers at the Universidade de Cantabria.

Also, you refer to GitHub several times in the Code and Data availability section. GitHub is not a suitable repository for scientific publication. GitHub itself instructs authors to use other alternatives for long-term archival and academic publishing, such as Zenodo (GitHub provides a direct way to copy your project to a Zenodo repository). Therefore, please, publish frozen versions of the code used in this work (this includes C4R) in one of the appropriate repositories, and remove from the text and notebooks the mentions to GitHub, as they can be confusing to the reader.

A note, the Terms of Use to access the User Data Gateway at the Universidad de Cantabria are not available. The link shows an ERROR 500 message.

*Response: We have followed the indications of the editor and both the data and code (notebook) are now available on Zenodo (the DOI is included in the revised manuscript). The notebook is now fully reproducible complying with the GMD code and data policy.*

**Paper gmd-2022-57: "Downscaling Multi-Model Climate Projection Ensembles with Deep Learning" by Baño-Medina et al.**

**Comments from Reviewer 1**

General comments: In this paper, the author built a statistical downscaling model based on (Convolutional Neural Network) CNN. And used the reanalysis data as a reference for the training phase. Then, the author employed the trained model to downscale the ensemble of CMIP5 models over Europe. The results were compared with the CORDEX RCMs for the historical (1975-2005) and the projection (2006-2100) period. The results from the deepSD algorithm are contributing to CORDEX initiative, which is a breakthrough since the statistical downscaling based on artificial intelligence was not trusted for climate studies in the past few years. In general terms, the manuscript is well structured, the methodology is well described, the figures and tables are well organized and the results are adequately discussed. This paper falls within the scope of this journal. In this sense, the article can be approved after minor revisions.

*Response: We thank the reviewer for his/her encouraging comments and sincerely appreciate the time taken to review the manuscript.*
*Please note that we have performed a substantial revision of the manuscript in response to the reviewers' comments. In particular, we have revised the adjustment procedure used to harmonize the GCM predictors, as suggested by some reviewers, adopting a change-preserving method (as suggested in the literature). This resulted in a reduced ensemble spread for temperature (comparable to the original GCM spread) allowing us to explain the main issue reported in the original manuscript.*

Abstract. The sentence "To our knowledge, this is the first time that CNNs have been used to produce multi-model ensembles" is not that accurate since there are previous studies that employed CNN to downscale the model ensemble (e.g., Babaousmail et al. (2021)).

*Response: We thank the reviewer for this interesting reference. We have noticed that in [1], the Deep Learning topology is used as a Model Output Statistics (MOS) technique, since it uses the precipitation —which is also the target variable— of the Global Climate Model (GCM) as input data. This differs from our approach which falls in the "perfect-prognosis" family within statistical downscaling, using large scale predictors from reanalysis data as input to calibrate the model. For this reason we have rephrased the sentence indicated by the reviewer to the following one: "To our knowledge, this is the first time that CNNs have been used to produce downscaled multi-model ensembles based on the perfect-prognosis approach".*
*[1] Babaousmail, Hassen, et al. "Novel statistical downscaling emulator for precipitation projections using deep Convolutional Autoencoder over Northern Africa." Journal of Atmospheric and Solar-Terrestrial Physics 218 (2021): 105614.*

Introduction (third paragraph): "v…". This sentence needs a citation.
*Response: Compared to regional climate models (RCMs), which require computation times of months for the simulations required in the CORDEX initiative, statistical models (including convolutional neural networks) can be produced in times of hours to days (as in this paper). We have included the following reference:*

*[1] Le Roux, Renan, et al. "Comparison of statistical and dynamical downscaling results from the WRF*

*model." Environmental modelling & software 100 (2018): 67-73.*

The author should justify why he selected the RCP8.5 scenario out of the other scenarios?
*Response: We have included the following sentence in the manuscript "We follow previous work in this field [1,2] and select the RCP8.5 scenario, which shows the strongest climate change signal (especially for temperature) and therefore allows the extrapolation capability of CNNs to be optimally explored."*

*[1] Baño-Medina, J., Manzanas, R. and Gutiérrez, J.M.. "On the suitability of deep convolutional neural networks for continental-wide downscaling of climate change projections." Climate Dynamics 57.11 (2021): 2941-2951.*
*[2] Olmo, M.E., Balmaceda-Huarte, R. and Bettolli, M.L. "Multi-model ensemble of statistically downscaled GCMs over southeastern South America: historical evaluation and future projections of daily precipitation with focus on extremes." Climate Dynamics (2022): 1-18.*

Also, was there any method employed for the selection of the 8 GCMs?
*Response: The main reason for selecting this set is that it has already been used in other studies, thus allowing for comparison. In addition, predictors are publicly available (allowing for reproducibility, as illustrated in the supplementary notebook) and have been assessed in a previous study [1] and have also been utilized in EURO-CORDEX to drive RCMs [2]. This has been clarified in the revised manuscript.*

*[1] Brands, S., et al. "How well do CMIP5 Earth System Models simulate present climate conditions in Europe and Africa?." Climate dynamics 41.3 (2013): 803-817.*
*[2] Vautard, Robert, et al. "Evaluation of the large EURO-CORDEX regional climate model ensemble." Journal of Geophysical Research: Atmospheres 126.17 (2021): e2019JD032344.*

The author didn't justify why E-OBS v20 was selected as an observation in this study.
*Response: We have added the following paragraph to the manuscript:*
*"E-OBS is a high-resolution observational dataset generated by spatially interpolating the European Climate Assessment & Dataset (ECA&D) network of stations [1]. Although national and sub-national datasets exist, E-OBS accurately represents the regional climate over the entire European continent [2] and it is commonly used in continental-wide statistical downscaling experiments [3,4,5,6]. We chose version 20 (v20, release date October 2019) since it was the most recent at the start of this study."*

*[1] Klok, E. J., and A. M. G. Klein Tank. "Updated and extended European dataset of daily climate observations." International Journal of Climatology: A Journal of the Royal Meteorological Society 29.8 (2009): 1182-1191.*
*[2] Bandhauer, Moritz, et al. "Evaluation of daily precipitation analyses in E-OBS (v19. 0e) and ERA5 by comparison to regional high-resolution datasets in European regions." International Journal of Climatology 42.2 (2022): 727-747.*
*[3] Maraun, Douglas, et al. "VALUE: A framework to validate downscaling approaches for climate change studies." Earth's Future 3.1 (2015): 1-14.*
*[4] Vrac, Mathieu, and Pradeebane Vaittinada Ayar. "Influence of bias correcting predictors on statistical downscaling models." Journal of Applied Meteorology and Climatology 56.1 (2017): 5-26.*
*[5] Baño-Medina, J., Manzanas, R. and Gutiérrez, J.M. "Configuration and intercomparison of deep learning neural models for statistical downscaling." Geoscientific Model Development 13.4 (2020): 2109-2124.*
*[6] Baño-Medina, Jorge, Rodrigo Manzanas, and José Manuel Gutiérrez. "On the suitability of deep*

*convolutional neural networks for continental-wide downscaling of climate change projections." Climate Dynamics 57.11 (2021): 2941-2951.*

Data and Methods. Since the author is comparing the ensemble resulting from 8 GCMs with the RCM ensemble projection, shouldn't be the number of GCMs equal to the number of RCMs?

*Response:* As the reviewer mentions, it would be ideal to have the same number of members in the GCM and RCM sets. Under this assumption, there would only be 1 RCM per GCM, resulting in a total of 8 members for both sets. However, we wanted to avoid possible artifacts in the results due to the lack of variability in RCM selection. For this reason we occasionally used 2 RCMs per GCM, representing a compromise between having a similar - but not equal - number of members for each ensemble and partially including this source of uncertainty in the results.

Usually, when we train a neural net model, a validation phase is required after the training and it should be selected from the historical 25 years period, in this paper the author didn't mention it.

*Response:* During training, we use a test set (10% of the data) to perform early-stopping as in [1] (the training stops when the test error starts increasing). We have included the following phrase to mention this aspect: "During calibration, we use a test set (randomly selected 10% of the data) to perform early-stopping, and stop training when the test error stops decreasing after 30 epochs". Note that a cross-validation of the model (using reanalysis predictors) was performed in the previous paper [1]. This has been mentioned in the revised manuscript.

[1] *Baño-Medina, J., Manzanas, R. and Gutiérrez, J.M. "Configuration and intercomparison of deep learning neural models for statistical downscaling." Geoscientific Model Development 13.4 (2020): 2109-2124.*

Concerning the CNN algorithm, we noticed that the CNN used to downscale precipitation has one more layer than the one for temperature (one output layer). Can the author explain the reason?

*Response:* The configuration of the DeepESD method was undertaken in [1] and here we used the optimum configurations found in that study (one and ten kernels in the last convolutional hidden layer for precipitation and temperature, respectively). The number of output neurons is different in both networks (2*n for temperature and 3*n for precipitation, where n is the number of gridboxes), in order to represent the statistical parameters of the parametric distributions estimated per gridbox. For temperature, Gaussian daily conditional distributions are used (given by 2 parameters: mean and standard deviation), while for precipitation t we used Bernoulli-Gamma distributions (given by three parameters: probability of rain, shape and scale). More details can be found in the reference describing the convolutional network used [1], cited in the manuscript.

[1] *Baño-Medina, J., Manzanas, R. and Gutiérrez, J.M. "Configuration and intercomparison of deep learning neural models for statistical downscaling." Geoscientific Model Development 13.4 (2020): 2109-2124.*

**Paper gmd-2022-57: "Downscaling Multi-Model Climate Projection Ensembles with Deep Learning" by Baño-Medina et al.**

**Comments from Reviewer 2**

The manuscript entitled "Downscaling Multi-Model Climate Projection Ensembles with Deep Learning (DeepESD): Contribution to CORDEX EUR-44" applied a Convolutional Neural Networks (CNN) to downscale present-day and future GCM simulations, with a focus on precipitation and temperature simulation. The paper falls within the scope of the journal. The manuscript is well structured, the method is generally clearly presented, and the results fully support its conclusion. I have a few specific comments regarding the method and analyses.

*Response: We thank the reviewer for his/her encouraging comments and sincerely appreciate the time taken to review the manuscript.*

*Please note that we have performed a substantial revision of the manuscript in response to the reviewers' comments. In particular, we have revised the adjustment procedure used to harmonize the GCM predictors, as suggested by some reviewers, adopting a change-preserving method (as suggested in the literature). This resulted in a reduced ensemble spread for temperature (comparable to the original GCM spread) allowing us to explain the main issue reported in the original manuscript.*

One common problem I found in the manuscript is the use of specific terms that might be well known in the Deep Learning area but is unfamiliar to me, a GCM and RCM modeler, and people like me. I suppose the authors may want to broaden their impact not only in the DP but also on people working on dynamic modeling. Below I have listed a few:

Line 32: "perfect prognosis" please explain the term in detail

*Response: We have extended the concept of "perfect-prognosis" in the introduction as follows: "Under the "perfect-prognosis" (PP) approach, the statistical models are trained to learn a predictor-predictand link in an historical period using simultaneous observed and reanalysis (quasi-observations) values (daily in this work) for predictands and predictors, respectively. The resulting models are then applied to GCM predictor values (from present climate or future scenarios) to obtain the regional/local downscaled results. This approach is based on a number of assumptions. For example, predictors have to be realistically simulated by the GCMs (e.g. exhibiting small systematic biases), so large-scale fields in upper levels (less affected by orography and model resolution) are typically used as predictors (perfect prognosis assumption); moreover, the statistical models trained in present climate conditions should remain valid under modified (out-of-sample) climate conditions (generalization assumption) (see [1] for more details)."*

*[1] Gutiérrez, J.M. et al. "An intercomparison of a large ensemble of statistical downscaling methods over Europe: Results from the VALUE perfect predictor cross‐validation experiment." International journal of climatology 39.9 (2019): 3750-3785.*

Lines 35-36: please include 1-2 sentences to introduce dynamic downscale as a comparison to statistical downscale.

*Response: We have included the following sentence: " Compared to dynamical downscaling, ESD lacks explicit physics in the model formulation and typically does not ensure full multivariate (intervariate and spatial) consistency. However, these methods overcome the systematic biases present in RCM products (as the model is trained using*

*observations) and are not computationally demanding, avoiding the need for large computational infrastructures [1]. Therefore, these methods could be widely used to downscale global multi-model ensembles providing downscaled results at continental scales, e.g. in CORDEX domains."*

*[1] Le Roux, Renan, et al. "Comparison of statistical and dynamical downscaling results from the WRF model." Environmental modelling & software 100 (2018): 67-73.*

Line 61: Please provide more information about "E-OBS v20"
*Response: We have added the following paragraph to the manuscript:*
*"E-OBS is a high-resolution observational dataset generated by spatially interpolating the European Climate Assessment & Dataset (ECA&D) network of stations [1]. Although national and sub-national datasets exist, E-OBS accurately represents the regional climate over the entire European continent [2] and it is commonly used in continental-wide statistical downscaling experiments [3,4,5,6]. We chose version 20 (v20, release date October 2019) since it was the most recent at the start of this study."*

*[1] Klok, E. J., and A. M. G. Klein Tank. "Updated and extended European dataset of daily climate observations." International Journal of Climatology: A Journal of the Royal Meteorological Society 29.8 (2009): 1182-1191.*
*[2] Bandhauer, Moritz, et al. "Evaluation of daily precipitation analyses in E‐OBS (v19. 0e) and ERA5 by comparison to regional high‐resolution datasets in European regions." International Journal of Climatology 42.2 (2022): 727-747.*
*[3] Maraun, Douglas, et al. "VALUE: A framework to validate downscaling approaches for climate change studies." Earth's Future 3.1 (2015): 1-14.*
*[4] Vrac, Mathieu, and Pradeebane Vaittinada Ayar. "Influence of bias correcting predictors on statistical downscaling models." Journal of Applied Meteorology and Climatology 56.1 (2017): 5-26.*
*[5] Baño-Medina, Jorge, Rodrigo Manzanas, and José Manuel Gutiérrez. "Configuration and intercomparison of deep learning neural models for statistical downscaling." Geoscientific Model Development 13.4 (2020): 2109-2124.*
*[6] Baño-Medina, Jorge, Rodrigo Manzanas, and José Manuel Gutiérrez. "On the suitability of deep convolutional neural networks for continental-wide downscaling of climate change projections." Climate Dynamics 57.11 (2021): 2941-2951.*

Line 62: I assume "—" is a typo?
*Response: Yes, thank you for noticing it.*

Line 70: Can you explain the "harmonization process" further?
*Response: The harmonization process has been clarified in the revised manuscript (data and methods section), including a new illustrative figure illustrating the process.*

Lines 75-77: The method of DL should be explained in further detail. "They consist of … (one per each gridpoint in E-OBS)". I found many terms in these sentences that might be barriers to fully understand the method. Can you rephrase it?
*Response: We have changed the explanation of the CNN topology clarifying the concepts and citing the appropriate references. "In particular, we deploy the best performing topologies developed in [1], a recent study which intercompares different CNNs over Europe to downscale temperature (precipitation). They consist of three convolutional layers [2] with 50, 25 and 10 (1) spatial kernels (3x3 gridboxes) followed by a dense connection linking the*

*last hidden layer to the output neurons (corresponding to the land gridpoints in E-OBS). As in [1] we apply a distributional downscaling approach and use the network to estimate daily predictor-conditioned Gaussian (Bernoulli-Gamma) distributions for temperature (precipitation). This is implemented for each land gridbox using two (three) output neurons corresponding to the distributional parameters: mean and variance (probability of rain, shape and scale factors). The resulting networks are trained to optimize the negative log-likelihood of the Gaussian (Bernoulli-Gamma) distribution. We refer the reader to [1] for more details."*

*[1] Baño-Medina, Jorge, Rodrigo Manzanas, and José Manuel Gutiérrez. "Configuration and intercomparison of deep learning neural models for statistical downscaling." Geoscientific Model Development 13.4 (2020): 2109-2124.*
[2] LeCun, Yann, and Yoshua Bengio. "Convolutional networks for images, speech, and time series." *The handbook of brain theory and neural networks* 3361.10 (1995): 1995.

Line 97-98: Merge the single sentence to the following paragraph
***Response:*** *Done.*

Line 120: It is interesting that the DeepESD has the smallest ensemble spread over the historical period (Fig. 2) but has the largest one over the future. Any explanation for that?
***Response:*** ESD methods achieve (by construction) a reduction of the model biases in the historical period, thus leading to a small ensemble spread in the temperature/precipitation output fields. However, when looking at future periods the ensemble spread of the downscaled climate change signal is the result of 1) the different trends of the predictor fields across GCMs, and 2) the extrapolation ability of the downscaling method. As described in the initial response, the adoption of a new change preserving adjustment method has resulted in comparable spread of row and downscaled ensembles for temperature over the future (and reduced spread for precipitation).

125: what do "these differences" refer to?
***Response:*** It refers to the differences in the climate change signal between the GCM and the RCM/DeepESD, which are described in the preceding paragraph. This has been clarified in the revised manuscript.

Line 130: "These differences are quite systematic for the case of precipitation indicating a robust CNN extrapolation fingerprint." Can you explain in further detail?
***Response:*** Thank you for this comment. We agree that this sentence may be misleading. However, we have changed Figure 4 in the revised manuscript and this sentence is no longer applicable.

Figure 4: How do you produce Row 3? Is it the difference (DeepESD minus E-OBS v2.0) shown in each model in row 2 minus the mean of the difference? In that case, the mean of 8 panels in row 3 should be 0, but clearly they are not.
***Response:*** *Figure 4 has been modified in the revised manuscript by deleting row 3 and including both the GCM and DeepESD downscaled climate change signals (as well as their differences). We believe this helps to clarify the figure and better convey the results.*

**Paper gmd-2022-57: "Downscaling Multi-Model Climate Projection Ensembles with Deep Learning" by Baño-Medina et al.**

**Comments from Reviewer 3**

General comments: Recently, the rapid development of deep learning (DL) technologies has provided novel approaches to construct empirical-statistical models for downscaling GCM outputs. DL can identify and extract complex nonlinear relationships that exist in large data sets and model them to develop the downscaling technique. There are progressively more studies exploring a proof of the concept for developing empirical-statistical models based on the DL technique, as in this work.

In this study, the deep learning algorithm of convolutional neural networks (CNN) is used as a statistical downscaling technique. The CNN-based model was trained using daily predictors from the ERA-Interim reanalysis and predictands from E-OBS v20 observations. And then, the well-trained model was applied to downscale eight GCMs outputs for the historical and RCP8.5 periods. The downscaled projections by the CNN-based model were compared with results from RCMs driven by the same GCMs. The authors found that the CNN-based model can provide comparable climate changes signals to those obtained with the RCMs, with a smaller uncertainty for precipitation.

This study shows promising results and the manuscript is well written. However, before being accepted, I think the manuscript could be improved as commented below. I believe my comments/suggestions are not very critical and do not require hard work, and this manuscript would be suitable for publication after appropriate revision.

*Response: We thank the reviewer for his/her encouraging comments and sincerely appreciate the time taken to review the manuscript.*

*Please note that we have performed a substantial revision of the manuscript in response to the reviewers' comments. In particular, we have revised the adjustment procedure used to harmonize the GCM predictors, as suggested by some reviewers, adopting a change-preserving method (as suggested in the literature). This resulted in a reduced ensemble spread for temperature (comparable to the original GCM spread) allowing us to explain the main issue reported in the original manuscript.*

As RCMs require massive computational resources, the CNN-based model may be plausible to be used as an alternative for downscaling GCM outputs. So, please add some discussions about the computational efficiency of the CNN-based model.

*Regarding the computational resources, we have included some references comparing dynamical and statistical downscaling [1] and have added the computational resources and time required for our model for the experiment described in the manuscript: "The computations performed in this work were executed on a single node 2x Intel(R) Xeon(R) E5-2670 0 @ 2.60GHz CPU (16 cores) with 60 GiB of RAM. The computational time taken to calibrate the model and generate the projections for a GCM was less than six hours, which is considerably less than the time required to run a similar experiment with an RCM (for instance, the EUR-44 simulations performed with the WRF model for a single GCM in [2] lasted six months using 9 nodes with 144 cores)."*

*[1] Le Roux, Renan, et al. "Comparison of statistical and dynamical downscaling results from the WRF model." Environmental modelling & software 100 (2018): 67-73.*
*[2] Fernández, J., Frías, M.D., Cabos, W.D. et al. Consistency of climate change projections from*

*multiple global and regional model intercomparison projects. Clim Dyn 52, 1139–1156 (2019). https://doi.org/10.1007/s00382-018-4181-8*

Line 12: "… but a similar uncertainty for temperature". More precisely, it is a larger uncertainty for temperature, as shown in Figure 3.
*Response:* Yes, thank you for noticing it. However, based on the comments of some reviewers asking to explore the influence of the harmonization process in the downscaling model outputs, we have rebuilt DeepESD using a change-preserving adjustment technique. The new results achieve indeed "a similar uncertainty for temperature" even reducing it with respect to the GCM one in some regions. Therefore we have rephrased Line 12 to "DeepESD preserves the uncertainty for temperature and results in a reduced uncertainty for precipitation".

Line 71−73: The CNN-based model provides smaller uncertainties for precipitation. Does the harmonization process contribute to this respect? Please add discussions about the role of the harmonization process.
*Response: The harmonization process has been clarified in the revised manuscript (data and methods section), including a new illustrative figure illustrating the process. Please note that we have revised the adjustment procedure used to harmonize the GCM predictors, as suggested by some reviewers, adopting a change-preserving method (as suggested in the literature). This resulted in a reduced ensemble spread for temperature (now comparable to the original GCM spread).*

Figure 1. Please add units (°C) to the color bar of the temperature bias.
*Response:* Done. Thank you for noticing it.

Line 102. The CNN-based model is trained with the observations. So it can be expected that DeepESD exhibits a largely unbiased spatial pattern. The CNN-based model is also used to downscale the GCM outputs for the historical period. How about the downscaled historical GCM simulations by the CNN-based model? Please add discussions about the bias of the CNN-based model for the historical period.
*Response:* Indeed, the bias pattern displayed in Figure 1, row 2, columns 3 and 6, is the bias (relative to E-OBS) of the downscaled GCM outputs for the historical period obtained with DeepESD. We have clarified this aspect by changing the caption as follows: "Annual daily precipitation (left block) and temperature (right) for the historical period 1975-2005, as obtained from the ensembles of GCMs, RCMs and DeepESD GCM-downscaled results (left, middle and right columns, respectively). The first row shows the ensemble mean climatological values and the second row displays the corresponding biases with respect to E-OBS v20."
Validation results with reanalysis (ERA-Interim) predictors are included in the original paper [1] showing mostly unbiased cross-validated patterns for the downscaled temperature and precipitation.

[1] *Baño-Medina, Jorge, Rodrigo Manzanas, and José Manuel Gutiérrez. "Configuration and intercomparison of deep learning neural models for statistical downscaling." Geoscientific Model Development 13.4 (2020): 2109-2124.*

L106. Figure 2 shows that DeepESD shows good performance in reproducing the variability

and extremes. I think this capability is important. Please add more discussions about Figure 2.

*Response:* We have added the following paragraph in the manuscript:

"Besides these results for the mean, Figure 3 compares the entire precipitation and temperature distributions for the GCM, RCM and DeepESD ensembles over the historical period 1979-2005, for three different illustrative regions (the Alps, Iberian Peninsula and Eastern Europe). The reduction of biases is also noticeable along the entire distribution (including the extremes) for both precipitation and temperature. Note that for precipitation these results are due to the use of the stochastic nature of the method, sampling from the inferred conditional distributions. RCMs and, particularly, GCMs overestimate low rainfall events and underestimate the high rainfall ones. "

L114−116: I do not quite understand this sentence. Could you please explain it?

*Response: We indicate that for precipitation, the climate change signal and the inter-model uncertainty of DeepESD are more similar to the RCM than to the GCM fields (see Figure 4 in the revised version of the manuscript).*

**Paper gmd-2022-57: "Downscaling Multi-Model Climate Projection Ensembles with Deep Learning" by Baño-Medina et al.**

**Comments from Reviewer 4**

General comments: The paper is an important contribution to the methodology of the downscaling of the climate change simulations. The results show that neural networks can bring meaningful regionalized climate change fields that can be a good complement to those obtained from regional climate models. The method works well for precipitation and temperature, but question of its usefulness for other climate fields, specially 3d fields, remains open.

The paper is clearly written and well structured. However, there are a few things that can be improved in the paper. In particular, I think that the answer to these comments could improve the paper.

*Response: We thank the reviewer for his/her encouraging comments and sincerely appreciate the time taken to review the manuscript.*

*Please note that we have performed a substantial revision of the manuscript in response to the reviewers' comments. In particular, we have revised the adjustment procedure used to harmonize the GCM predictors, as suggested by some reviewers, adopting a change-preserving method (as suggested in the literature). This resulted in a reduced ensemble spread for temperature (comparable to the original GCM spread) allowing us to explain the main issue reported in the original manuscript.*

With which criteria were chosen the predictand fields?

*Response:* We chose to downscale air surface temperature and daily total precipitation since they are two variables highly demanded by the climate, impact and adaptation communities. These are also the variables with best observations available and, therefore, provide reliable training data sets. It would be interesting to extend this study to other relevant predictand variables (e.g., wind) and we leave this possible continuation to future work.

How expensive in computer resources is the method?

*Response:* In the revised manuscript we have included some general comments and references regarding computational requirements of dynamical and statistical downscaling (as requested by some reviewers). Besides, we have included the following results regarding the computational resources required by the method:

"*The computations performed in this work were executed on a single node 2x Intel(R) Xeon(R) E5-2670 0 @ 2.60GHz CPU (16 cores) with 60 GiB of RAM. The computational time taken to calibrate the model and generate the projections for a GCM was less than six hours, which is considerably less than the time required to run a similar experiment with an RCM (for instance, the EUR-44 simulations performed with the WRF model for a single GCM in [2] lasted six months using 9 nodes with 144 cores).*"

Why eobs was used? It is too smooth, what can be seen in the results, specially in places with high topography. Why did not used a regional, high resolution reanalysis as predictands?

*Response: We have added the following paragraph to the manuscript:*

*"E-OBS is a high-resolution observational dataset generated by spatially interpolating the European Climate Assessment & Dataset (ECA&D) network of stations [1]. Although national and sub-national datasets exist, E-OBS accurately represents the regional climate over the entire European continent [2] and it is commonly used in continental-wide statistical downscaling experiments [3,4,5,6]. We chose version 20 (v20, release date October 2019) since it was the most recent at the start of this study."*

*[1] Klok, E. J., and A. M. G. Klein Tank. "Updated and extended European dataset of daily climate observations." International Journal of Climatology: A Journal of the Royal Meteorological Society 29.8 (2009): 1182-1191.*

*[2] Bandhauer, Moritz, et al. "Evaluation of daily precipitation analyses in E‑OBS (v19. 0e) and ERA5 by comparison to regional high‑resolution datasets in European regions." International Journal of Climatology 42.2 (2022): 727-747.*

*[3] Maraun, Douglas, et al. "VALUE: A framework to validate downscaling approaches for climate change studies." Earth's Future 3.1 (2015): 1-14.*

*[4] Vrac, Mathieu, and Pradeebane Vaittinada Ayar. "Influence of bias correcting predictors on statistical downscaling models." Journal of Applied Meteorology and Climatology 56.1 (2017): 5-26.*

*[5] Baño-Medina, Jorge, Rodrigo Manzanas, and José Manuel Gutiérrez. "Configuration and intercomparison of deep learning neural models for statistical downscaling." Geoscientific Model Development 13.4 (2020): 2109-2124.*

*[6] Baño-Medina, Jorge, Rodrigo Manzanas, and José Manuel Gutiérrez. "On the suitability of deep convolutional neural networks for continental-wide downscaling of climate change projections." Climate Dynamics 57.11 (2021): 2941-2951.*

It seems that the use of more output layers for the precipitation than in temperature makes the biases in the downscaling of precipitation as small as for temperature, but reduces the standard deviation in the downscaling (Figure 3). I think that this fact is related to the methodology and should be commented on by the authors.

***Response:*** Both temperature and precipitation topologies have the same number of filter maps and hidden layers in their topologies (see the extended description of the methods introduced in the revised manuscript) and, thus, they can achieve the same degree of nonlinearity. The CNNs deployed contain different number of, n, output neurons (2*n for temperature and 3*n for precipitation), representing each of the statistical parameters of the parametric distributions estimated per gridbox. For temperature, we learn Gaussian daily conditional distributions parameterized by 2 parameters (mean and standard deviation) per predictand site, while for precipitation there are 3 parameters (probability of rain, shape and scale factor) corresponding to the Bernoulli-Gamma distribution. Having more output layers does not add non-linearity to the network, and therefore this aspect does not have an influence on the future estimates and indices —either biases or standard deviation. For more details on the topology we refer to [1].

*[1] Baño-Medina, Jorge, Rodrigo Manzanas, and José Manuel Gutiérrez. "Configuration and intercomparison of deep learning neural models for statistical downscaling." Geoscientific Model Development 13.4 (2020): 2109-2124.*

Also, the fact that the simulation of R01 in DeepESD is closer to the RCMs that to the GCMs shows the importance of a good simulation of orographic precipitation, while SDII and Mean temperature in DeepESD and GCM are closer, probably reflecting the tuning of the GCMs (which usually is not made in RCMs) and the training with observations in DeepESD. The

exception for temperature in ED looks strange for me and would be nice if you explain this behavior.

**Response:** This aspect is related to what was explained in lines 126-127: "In the case of the RCMs, some recent studies attribute these differences to the lack of time-varying anthropogenic aerosols in the RCM formulation (Boé et al., 2020; Gutiérrez et al., 2020)". Therefore, there is an on-going analysis by the dynamical downscaling community to analyze the differences mentioned by the reviewer in Eastern and Central Europe of the climate change signal of temperature between the GCMs and RCMs, investigating whether it is due to an added value of dynamical downscaling or to deficiencies in the model formulation of the RCMs.

Results in figures 3,4 and could be also contributed by the use of stochastic (deterministic) approaches for the precipitation (temperature) specific comments

**Response:** The choice for either stochastic or deterministic downscaled fields is mainly relevant to the reproduction of extremes (Figure 2), but they do have a negligible influence on the results of Figure 3 and 4, which display only the mean of temperature and precipitation. This comparison among stochastic and deterministic fields can be found in [1].

[1] *Baño-Medina, J., Manzanas, R. and Gutiérrez, J.M.. "On the suitability of deep convolutional neural networks for continental-wide downscaling of climate change projections." Climate Dynamics 57.11 (2021): 2941-2951.*

A more detailed description of the methodology for not specialists (most readers, I guess) should be interesting. Can be added as an appendix

**Response:** *We have changed the explanation of the CNN topology clarifying the concepts and citing the appropriate references. "In particular, we deploy the best performing topologies developed in [1], a recent study which intercompares different CNNs over Europe to downscale temperature (precipitation). They consist of three convolutional layers [2] with 50, 25 and 10 (1) spatial kernels (3x3 gridboxes) followed by a dense connection linking the last hidden layer to the output neurons (corresponding to the land gridpoints in E-OBS). As in [1] we apply a distributional downscaling approach and use the network to estimate daily predictor-conditioned Gaussian (Bernoulli-Gamma) distributions for temperature (precipitation). This is implemented for each land gridbox using two (three) output neurons corresponding to the distributional parameters: mean and variance (probability of rain, shape and scale factors). The resulting networks are trained to optimize the negative log-likelihood of the Gaussian (Bernoulli-Gamma) distribution. We refer the reader to [1] for more details."*

[1] *Baño-Medina, Jorge, Rodrigo Manzanas, and José Manuel Gutiérrez. "Configuration and intercomparison of deep learning neural models for statistical downscaling." Geoscientific Model Development 13.4 (2020): 2109-2124.*
[2] *LeCun, Yann, and Yoshua Bengio. "Convolutional networks for images, speech, and time series." The handbook of brain theory and neural networks 3361.10 (1995): 1995.*

How does the interpolation method influence the results?

**Response:** We tested two different approaches to re-grid the GCM predictor fields to a common 2° latitude-longitude: nearest-neighbour and bilinear interpolation. We found no remarkable differences in the climate change signals obtained for these two interpolation methods. We added a comment on this to the revised manuscript: "No differences in the downscaled fields were found by employing other interpolation techniques (e.g., bilinear)".

In the Iberian Peninsula and the Scandinavian peninsula the climate change signal in DeepESD is similar to that of the global models, while the opposite is true in central Europe. Could you elaborate on this?

*Response:*  We agree with the reviewer that this aspect is very interesting. Our plan (as included in the responses to the online discussions) was to explore in detail these differences in a future work. However, based on the interest of some reviewers in this aspect we have further explored these climate change signals during the reviewing stage of this manuscript and included the results in the revised version. We have found that the differences found between the climate change signals of the GCMs and DeepESD mentioned in you comment were a consequence of the harmonization process employed to adjust the GCM predictors (simple bias adjustment of the mean and variance), which changed the future changes of the predictors and thus also of the downscaled results. Therefore, we have changed the harmonization process following a change preserving approach, as suggested in [1], preserving the climate change signal of the predictors. The result is a closer resemblance of the GCM and DeepESD climate change spatial patterns for temperature. These results have been included in the revised manuscript, affecting mainly figures 3 and 4 and the above mentioned results.

[1] Vrac, M. and Ayar, P.: Influence of Bias Correcting Predictors on Statistical Downscaling Models, Journal of Applied Meteorology and Climatology, 56, 2016.

**Paper gmd-2022-57: "Downscaling Multi-Model Climate Projection Ensembles with Deep Learning" by Baño-Medina et al.**

**Comments from Reviewer 5**

In this study, the authors develop a downscaling method for climate variables based on the prefect prognosis approach using convolution neural networks (CNN) and evaluate how it extrapolates to unseen climatic states as projected by multi-model Earth system models. They focus on temperature and precipitation as well as on the European domain. The authors compare the output of their method to regional high-resolved climate model output and show that the CNN-approach reduces biases for the historical period and extrapolates to future climate change conditions in a plausible way.

The downscaling approach is very interesting and useful, especially the evaluation of its extrapolating skill to a different system state. Overall, the manuscript is well written. I have a few general comments and a short list of specific comments. Thus, I recommend minor revisions before publication.
*Response: We thank the reviewer for his/her encouraging comments and sincerely appreciate the time taken to review the manuscript.*
*Please note that we have performed a substantial revision of the manuscript in response to the reviewers' comments. In particular, we have revised the adjustment procedure used to harmonize the GCM predictors, as suggested by some reviewers, adopting a change-preserving method (as suggested in the literature). This resulted in a reduced ensemble spread for temperature (comparable to the original GCM spread) allowing us to explain the main issue reported in the original manuscript.*

Could you make flowchart of your workflow and include as a figure in the methods section? It is a bit hard to follow your exact procedure.
*Response: Thank you for this suggestion. We have included a new illustrative figure illustrating the process.*

Isn't the comparison to observations between unconstrained mechanistic models (i.e. GCMs) and CNNs trained on observations "unfair"? If you did some nudging procedure with GCMs you would also end up with model output better fitting observations. For the CNN training, did you split the observational data into train (validation) and test set (only train on 20 years and show performance for 10 years)? Again, a flowchart would help to understand what you did. If you show the performance of DeepESD for the test set and compare that to GCM output, it'd be "more fair", but still, just by design we would expect that the CNN reproduces observations better than GCMs.
*Response: The flowchart indicated in the previous comment probably sheds light on this one as well. Indeed, the train and validation sets differ in the predictor datasets (ERA-Interim for training and GCMs for validation) rather than in the temporal period. Cross-validation results using ERA-Interim predictors are provided in [1] and cited in the paper. As the reviewer points out, the comparison between GCM and DeepESD is still not totally "fair" in the historical period. However, in the manuscript we do not intend to establish an argument in this line, but rather to compare the climate change signals between ensembles whilst showing DeepESD a good reproducibility of the local scale in the historical period.*

[1] Baño-Medina, J., Manzanas, R. and Gutiérrez, J.M. "Configuration and intercomparison of deep learning neural models for statistical downscaling." Geoscientific Model Development 13.4 (2020): 2109-2124.

You show that the CNN learns the "necessary" dynamics based on predictors of the historical period and extrapolates reasonably well using predictors from GCM output for projections. That is a very interesting point. I wonder if this simple bias correction for GCMs really does the trick, as the models considerably diverge over the climatic time-scales and very model specific regional biases emerge. Can you comment on whether other bias-correcting measures were tested? Overall, there is certainly a long list of potential further evaluation and testing steps that could be undertaken, but maybe it is enough for this model description paper.

*Response: The harmonization process has been clarified in the revised manuscript (data and methods section), including a new illustrative figure illustrating the process. Please note that we have revised the adjustment procedure used to harmonize the GCM predictors, as suggested by some reviewers, adopting a change-preserving method (as suggested in the literature). This resulted in a reduced ensemble spread for temperature (now comparable to the original GCM spread).*

L5: What is DeepESD standing for? Please introduce acronym before first usage.
*Response: It stands for Deep learning Empirical Statistical Downscaling (DeepESD). We have introduced this acronym in the manuscript.*

LL33-34: The "perfect prognosis" approach is based on the assumption that GCMs don't have systematic biases with respect to the observations that were used for training, right? Maybe you should include a short sentence here that addresses this aspect.
*Response: We have included a more detailed description of Perfect Prognosis in the introduction of the revised manuscript, including the above mentioned issue of biases.*

L55: I recommend to use another more static hosting platform for your code, e.g. Zenedo.
*Response: Both the code and the input data are hosted in Zenodo in the revised version.*

L60: Why did you use ERA-Interim reanalysis? It is outdated for quite some time now.
*Response: This study builds on previous ones which use ERA-Interim data to deploy CNNs over Europe [1,2]. For consistency with these studies and also with reference statistical downscaling experiments in the continent [3,4,5,6] which also build on ERA-Interim data, we decided to use this dataset for the predictors. However, we plan to move to ERA5 to downscale CMIP6 GCMs in future work.*

*[1] Baño-Medina, J., Manzanas, R. and Gutiérrez, J.M. "Configuration and intercomparison of deep learning neural models for statistical downscaling." Geoscientific Model Development 13.4 (2020): 2109-2124.*
*[2] Baño-Medina, J., Manzanas, R. and Gutiérrez, J.M. "On the suitability of deep convolutional neural networks for continental-wide downscaling of climate change projections." Climate Dynamics 57.11 (2021): 2941-2951.*
*[3] Bedia, J. et al. "Statistical downscaling with the downscaleR package (v3. 1.0): contribution to the VALUE intercomparison experiment." Geoscientific Model Development 13.3 (2020): 1711-1735.*
*[4] Maraun, D., et al. "VALUE: A framework to validate downscaling approaches for climate change studies." Earth's Future 3.1 (2015): 1-14.*
*[5] Maraun, D., M. Widmann, and J.M. Gutiérrez. "Statistical downscaling skill under present climate*

*conditions: A synthesis of the VALUE perfect predictor experiment." International Journal of Climatology 39.9 (2019): 3692-3703.*

*[6] Gutiérrez, José Manuel, et al. "An intercomparison of a large ensemble of statistical downscaling methods over Europe: Results from the VALUE perfect predictor cross‑validation experiment." International journal of climatology 39.9 (2019): 3750-3785.*

L62: I don't understand your use of dashes (—) in the manuscript. Please check whether the make sense throughout the manuscript.

*Response: The use of dashes has been avoided as much as possible in the revised manuscript.*

LL62-65: What about adding high-resolution orography description as static predictor?

*Response: This is done in other studies (e.g., super-resolution, where high-resolution predictors are used to downscale target variables) but is not a standard approach for perfect-prognosis downscaling.*

L85: Why did you analyze both and can you provide the reason why you settled with the deterministic one?

*Response: Sampling from the predicted conditional distributions permits reproducing the variability and extremes of the local time-series, which is reduced using the predicted expected value (deterministic method). However, since sampling is performed at each gridpoint individually, there is a loss in the spatial structure of the downscaled fields. Since local temperature is largely explained by the large-scale predictors, there is no need to sample from the inferred conditional distributions to preserve the variability. Conversely, local precipitation is not completely explained by the predictor set, and a stochastic downscaled version is needed to recover the variability of the target predictand. These aspects were already analyzed in a prior study [1] and for this reason we do not delve into too much detail in this manuscript.*

*[1] Baño-Medina, Jorge, Rodrigo Manzanas, and José Manuel Gutiérrez. "On the suitability of deep convolutional neural networks for continental-wide downscaling of climate change projections." Climate Dynamics 57.11 (2021): 2941-2951.*

L88: Please stick to the tenses (in this paragraph you mix present and past tense), i.e. do not switch between present and past tense when describing your results or methods. I recommend that you always use present tense when talking about your study, i.e. when describing your methods, your results etc., and use past tense when referring to already published studies.

*Response: Thank you.* Solved

L137: "contribute to increasing"

*Response: Thank you.* Solved

Figure 1: Add unit at lower right colorbar. Also, it'd be useful if you could include letter characters as pointers to subplots, e.g. a,b,c,d. This comment applies for all Figures.

*Response: We have included the units; thank you for picking this error. Regarding the subplot labels we analyzed this option but prefer to keep the figures as they are, since they can be easily explained using rows and columns.*

Figure 4: Please be more specific about the numbers in the plots. Please provide more detailed information in the caption.
*Response: Thank you.* Solved

Figure 5: The mid-column misses a time axis. DeepESD is not "yellow" but "green", no?
*Response: It is green. This typo has been corrected in the new version of the manuscript.*